# IL-12p35 induces expansion of IL-10 and IL-35-expressing regulatory B cells and ameliorates autoimmune disease

Ivy M. Dambuza[1], Chang He[1,2], Jin Kyeong Choi[1], Cheng-Rong Yu[1], Renxi Wang[1,3], Mary J. Mattapallil[4], Paul T. Wingfield[5], Rachel R. Caspi[4] & Charles E. Egwuagu[1]

Interleukin 35 (IL-35) is a heterodimeric cytokine composed of IL-12p35 and Ebi3 subunits. IL-35 suppresses autoimmune diseases while preventing host defense to infection and promoting tumor growth and metastasis by converting resting B and T cells into IL-10-producing and IL-35-producing regulatory B (Breg) and T (Treg) cells. Despite sharing the IL-12p35 subunit, IL-12 (IL-12p35/IL-12p40) promotes inflammatory responses whereas IL-35 (IL-12p35/Ebi3) induces regulatory responses, suggesting that IL-12p35 may have unknown intrinsic immune-regulatory functions regulated by its heterodimeric partner. Here we show that the IL-12p35 subunit has immunoregulatory functions hitherto attributed to IL-35. IL-12p35 suppresses lymphocyte proliferation, induces expansion of IL-10-expressing and IL-35-expressing B cells and ameliorates autoimmune uveitis in mice by antagonizing pathogenic Th17 responses. Recapitulation of essential immunosuppressive activities of IL-35 indicates that IL-12p35 may be utilized for in vivo expansion of Breg cells and autologous Breg cell immunotherapy. Furthermore, our uveitis data suggest that intrinsic immunoregulatory activities of other single chain IL-12 subunits might be exploited to treat other autoimmune diseases.

[1] Molecular Immunology Section, Laboratory of Immunology, National Eye Institute (NEI), National Institutes of Health (NIH), Bethesda, MD 20892, USA. [2] State Key Laboratory of Ophthalmology, Zhongshan Ophthalmic Center, Sun Yat-Sen University, Guangzhou 510060, China. [3] Laboratory of Immunology, Beijing Institute of Basic Medical Sciences, Beijing 100850, China. [4] Immunoregulation Section, Laboratory of Immunology, NEI, NIH, Bethesda, MD 20892, USA. [5] Protein Expression Laboratory, National Institute Arthritis and Musculoskeletal and Skin Diseases (NIAMS), National Institutes of Health, Bethesda, MD 20814, USA. Ivy M. Dambuza, Chang He and Jin Kyeong Choi contributed equally to this work. Correspondence and requests for materials should be addressed to C.E.E. (email: egwuaguc@nei.nih.gov)

The interleukin 12 (IL-12) family of cytokines (IL-12, IL-23, IL-27, and IL-35)[1–4] is known to consist of 4 members and each member is composed of two subunits, an α-subunit (IL-12p35, IL-23p19, and IL-27p28) and a β-subunit (IL-12p40, Ebi3)[1]. The subunits are each encoded by separate chromosomes and their expression is regulated independently[5]. The effects of IL-12 cytokines on host immunity derive from the fact that each of the α and β subunits is the target of microbial Toll-like receptor (TLR) agonists that activate innate immune cells, including monocytes and antigen-presenting dendritic cells[5]. Depending on the pathogen, activation of TLRs on the dendritic cells induces the transcription of distinct repertoires of the IL-12 α and β subunit genes[6–9]. The predominant IL-12 cytokine(s) produced within the immediate environment of differentiating naive lymphocytes is thought to influence the developmental decisions of the lymphocytes and thereby determines the lymphocyte subsets that would dominate the ensuing immune response. IL-12 family

cytokines are therefore considered to have critical functions in regulating the initiation, intensity, duration, and quality of adaptive immunity[1, 10, 11].

It is notable that the three α subunits are structurally related and each conceivably can pair with either of the structurally homologous β subunits[1, 10]. Pairing of the α-subunits, IL-12p35 or IL-23p19 with IL-12p40, gives rise to the two pro-inflammatory members IL-12 and IL-23, respectively, whereas the two immunosuppressive members of the family, IL-27 and IL-35, derive from pairing of IL-27p28 or IL-12p35 with Ebi3[1, 11]. In a previous study, we showed that IL-12p40-deficient mice are resistant to experimental autoimmune uveitis (EAU), suggesting that endogenous or IL-12 or IL-23 is required for induction and progression of EAU[12]. On the other hand, inflammatory stimuli induce microglial cells of the neuroretina to produce IL-27, and this immune-suppressive IL-12 member has been shown to inhibit Th17-mediated ocular inflammation and contribute to the

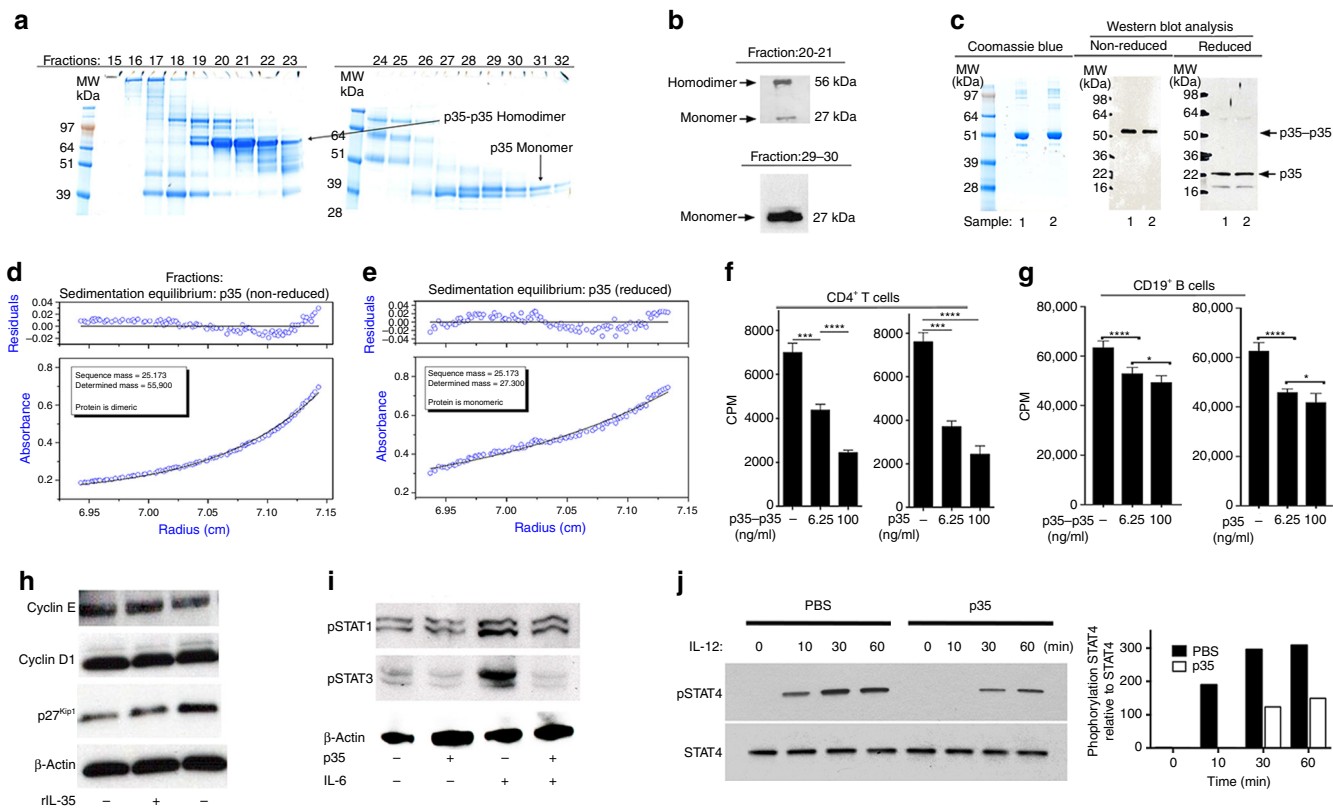

**Fig. 1** p35 monomer and homo-dimer suppressed T and B cell proliferation. Secreted rIL-12p35 was partially purified by His-tagged affinity chromatography using Ni-NTA followed by two consecutive cycles of gel filtration using Supercryl S-200 (HR Hiprep 16/60) and Superose-6 (HR 10/30) FPLC columns. **a** rIL-12p35 in insect cell supernatant was sequentially purified by Ni-NTA Purification system, size-exclusion Centricon filtration and Supercryl S-200 chromatography. Protein from Ni-chelate affinity column was applied to a Sephacryl S-200 column and the elution prolife screened by SDS-PAGE under non-reducing conditions is shown. Analysis of the FPLC column fractions by non-denaturing SDS-PAGE indicates presence of p35–p35 homo-dimer and p35 monomer in the rIL-12p35 preparation. **b** Fractions 20–21 (p35–p35) and 29–30 (p35) were fractionated under non-reducing condition on SDS-PAGE and analyzed by the western blotting. **c** Further purification on Superose-6 FPLC column. Analysis of the main peak by non-reducing or reducing SDS-PAGE; Data from Coomassie Blue stained gel and western blot analysis are indicated. Samples 1 and 2 are duplicates. **d, e** The peak protein fraction from Superose-6 column was further purified by size-exclusion Centricon filtration and subjected to sedimentation equilibrium analyses under non-reducing **d** or reducing **e** condition. The p35 is clearly dimeric in the absence of reductant and monomeric when reduced. **f** Naive CD4$^+$ T cells were stimulated with anti-CD3/CD28 in medium containing p35 or p35–p35 and after 72 h the effect of p35 or p35–p35 on proliferation was assessed by [$^3$H]-thymidine incorporation assay. **g** Purified CD19$^+$ B cells were stimulated with LPS for 2 days in medium containing p35–p35 or p35 and analyzed by [$^3$H]-thymidine incorporation assay. **h** Purified CD19$^+$ B cells were stimulated with LPS for 2 day in medium containing p35 or rIL-35 and analyzed by western blotting. **i, j** CD4$^+$ T cells were stimulated with anti-CD3/CD28 for 3 days. The cells were then cultured in serum-free medium for 2 h, followed by stimulation with IL-6 or IL-12 for 10, 30, and/or 60 min in the presence or absence of p35. Whole cell lysates were analyzed by western blotting using indicated Abs. Results represent at least three independent experiments and were analyzed using Student's $t$-test (two-tailed). Data are mean ± SEM. (***$P < 0.001$; ****$P < 0.0001$)

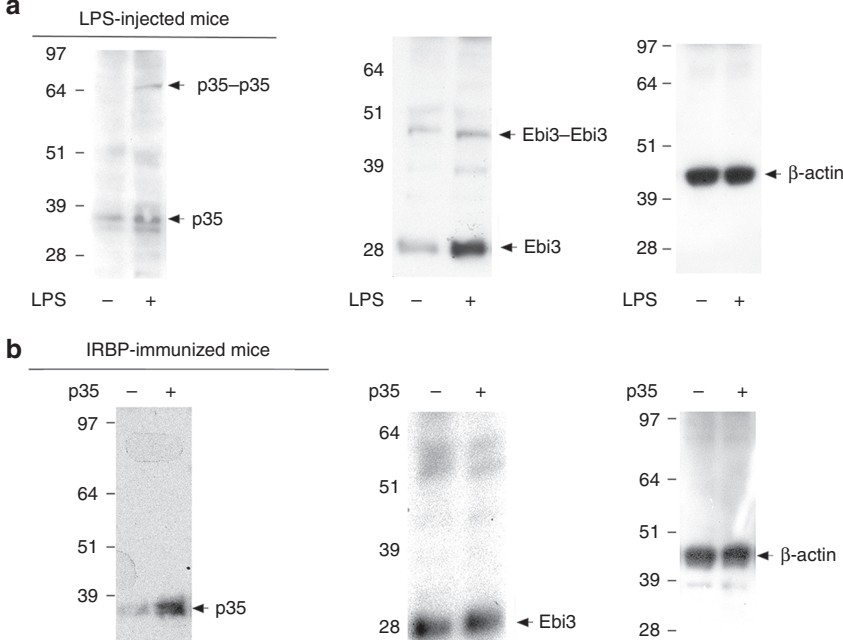

**Fig. 2** p35 induced an IL-35-producing Breg cells in vivo. **a** We injected C57BL/6 mice with LPS (15 µg/mouse) and after 4 days we isolated CD19⁺ B cells from the spleen (purity > 96%), lysed the cells and subjected the whole cell lysates to western blot analysis. **b** Intraocular inflammation (uveitis) was induced in C57BL/6J mice by immunization with the ocular autoantigen, IRBP in CFA as described in Methods section. Mice were killed 21 days post-immunization and cell lysate of the spleen was analyzed by western blotting (non-reduced condition). Results represent two independent experiments

maintenance of ocular immune privilege[13–15]. These and other reports underscore the emerging consensus that IL-12 and IL-23 are potential therapeutic targets that can be used to treat inflammatory diseases, whereas IL-27 and IL-35 are potential biologic agents for suppressing autoimmune diseases such as uveitis and multiple sclerosis. Aside from functioning as hetero-dimers, the individual subunits can also function autonomously as monomers or homo-dimers. For example, the IL-12p35 sub-unit has been shown to act as negative regulator of IL-27 responses in an experimental model of arthritis[16]. IL-27p28 also antagonizes IL-27 signaling, functioning as a natural antagonist of gp130-mediated signaling that can be exploited therapeutically to mitigate inflammatory diseases mediated by cytokines that utilize gp130[17]. On the other hand, IL-12p40 is secreted independently of IL-12 in serum of patients with pulmonary sarcoidosis and considered a useful clinical marker for disease activity in pul-monary sarcoidosis[18]. Similarly, IL-12p40 and disulfide-linked p40–p40 homo-dimer are secreted in serum of patients with multiple sclerosis and are associated with suppressing neurolo-gical dysfunctions or endotoxemia by antagonizing IL-12 sig-naling and Th1 expansion[19, 20]. Of relevance to the development of novel therapeutic cytokines/biologic agents is the proposition that altering the balance between the different IL-12 subunits may be a strategy to regulate inflammatory responses.

IL-35 is the other anti-inflammatory member of the IL-12 family of cytokines[21–23]. IL-35 is composed of Ebi3, a β-chain subunit encoded by the Epstein–Barr virus (EBV)-induced gene 3 (*Ebi3*, also known as *IL27b*), and the IL-12p35 α subunit encoded by *IL12a*[21, 22, 24]. Initial reports indicated that IL-35, produced mainly by T cell contributes to the suppressive activities of reg-ulatory T (Treg) cells[21]. However, subsequent reports have shown that IL-35 is also a physiological inducer of IL-10-producing regulatory B (Breg) cells, as well as, a relatively rare B cell sub-population that produces IL-35[25]. In addition, IL-35 confers protection of mice from uveitis or encephalitis by inhibiting Th17 and Th1 auto-reactive pathogenic T cells while promoting the

expansion of B and T cells[25, 26]. Despite interest in IL-35 as potential biologic agent for treatment of autoimmune diseases, the mechanism by which it mediates its biological effects is not fully understood. Although it is implicitly assumed that immu-nosuppressive activities of IL-35 derive from pairing of IL-12p35 and Ebi3, it is unknown whether the single chain proteins, IL-12p35 and Ebi3, possess intrinsic biological activities. More-over, the relative contribution of either chain to the immune-regulatory functions of IL-35 is unclear. In this study, we have produced mouse recombinant IL-12p35 (rIL-12p35) and rEbi3 and examined whether either protein can recapitulate some of the inhibitory activities of IL-35. Our data indicate that IL-12p35, while not readily detectable in vivo in the steady-state, possesses at least some of the immune-regulatory properties of the heterodimeric IL-35 cytokine and when used therapeutically it is able to control autoimmune disease affecting the neuroretina.

## Results

**IL-12p35 suppresses proliferation of T and B cells.** Biologically active, heterodimeric IL-35 (rIL-35) was previously produced by expressing the IL-12p35 and Ebi3 cDNA constructs in insect cells using the pMIB bicistronic vector[25]. In this study, we used the pMIB expression system to produce the mouse recombinant IL-12p35 (rIL-12p35) or Ebi3 (rEbi3) single chain protein; the powerful polyhedrosis Baculovirus early promoter drove expres-sion of the cDNA construct and secretion was directed by the honeybee melittin (HBM) signal peptide (Fig. 1 and Supple-mentary Fig. 1). The secreted rIL-12p35 migrates on non-denaturing (non-reduced) SDS-PAGE as a p35–p35 homo-dimer of ~57 kDa and a p35 monomer of ~27 kDa (Fig. 1a). Analyses of FPLC fractions 20–21 and 29–30 by western blotting confirmed that the secreted p35–p35 homo-dimer and the p35 monomer are indeed IL-12p35 (Fig. 1b). The partially purified proteins were subjected to additional purification on superpose-6 FPLC columns and by enrichment with Ultra centrifugal

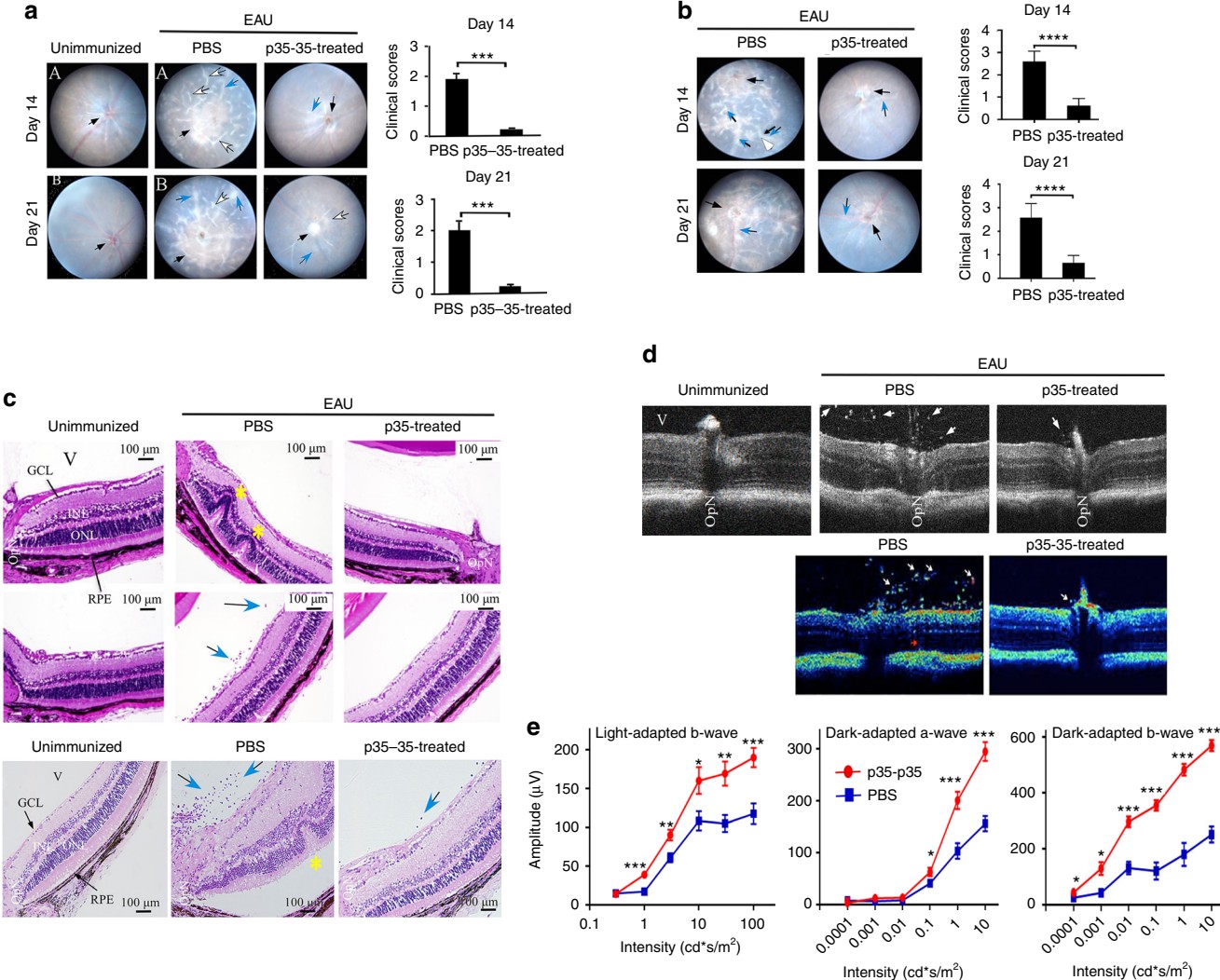

**Fig. 3** p35 and p35−p35 mitigated experimental autoimmune uveitis (EAU). EAU was induced in C57BL/6J mice and some mice were treated with p35−p35 or p35. Fundus images **a**, **b** and histology sections **c** of the retina at day 14 or day 21 after EAU induction. Fundus images were taken using an otoendoscopic imaging system. Note inflammation with blurred optic disc margins and enlarged juxtapapillary area (*black arrows*), retinal vasculitis with moderate cuffing (*blue arrows*), and *yellow−whitish* retinal and choroidal infiltrates (*white arrows*). Clinical scores and assessments of disease severity were based on pathological changes at the optic disc and retinal and choroidal tissues. H&E histological sections **c** show inflammatory cells in retina and vitreous (*blue arrows*) and numerous retinal folds, a hallmark of severe retinitis *yellow highlight*. OPN, optic nerve; GCL, ganglion cell layer; INL, inner nuclear layer; ONL, outer nuclear layer; RPE (retinal pigmented epithelial cell layer) and choroid. **d** Layered structure of the retina was visualized by SD-OCT. Representative OCT images taken on day 18 after disease induction show markedly increased inflammatory cells (*white arrows*) in the vitreous, retina and posterior chamber of the untreated mice compared to those treated with p35 or p35−p35. Retinal layer destructions (*red arrow*). **e** ERG analysis of the retina on day 20 after EAU induction. The averages of light-adapted or dark-adapted ERG b-wave amplitudes are plotted as a function of flash luminance and values are mean ± SEM from four animals in each group. Results represent at least three independent experiments and were analyzed using Student's *t*-test (two-tailed). (*$P < 0.05$; **$P < 0.01$; ***$P < 0.001$; ****$P < 0.0001$)

size-exclusion filters (Centricon). The purified p35 monomer and homo-dimeric (p35−p35) proteins used in all studies described here were verified to be IL-12p35 by western blot analysis using IL-12p35-specific antibodies, under reducing or non-reducing condition (Fig. 1c). The purified proteins (fractions 20–21 and 29–30) were also analyzed under native conditions by sedimentation equilibrium to directly determine their molecular weights. The results confirmed that rIL-12p35 can exist in solution as a homo-dimer (p35−p35) with a molecular weight of ~ 56 kDa (Fig. 1d) and predominantly as a monomer of ~ 27 kDa under reduced condition (Fig. 1e). While the rIL-12p35 preparation contained p35 and p35−p35 proteins, we cannot rule out possibility that presence of both moieties might be influenced by the oxidative state under in vivo conditions or

during purification of the secreted proteins. Assuming that affinity of hetero-dimerization might be stronger than homo-dimerization, it is conceivable that p35 does not accumulate in concentrations required for homo-dimerization in vivo, but would be mopped up forming heterodimers with Ebi3 (e.g., IL-35) or p40 (IL-12). Nonetheless, our in vitro studies show that either p35 or p35−p35 is able to suppress T cell (Fig. 1f) and B cell (Fig. 1g) proliferation in dose-dependent manner, establishing that the monomer and homodimer are both biologically active. IL-35 was previously shown to inhibit lymphocyte proliferation by upregulating p27[kip1] and thereby restricting progression through the S phase of the cell cycle[27, 28]. We therefore activated CD19[+] cells with LPS in medium containing p35 or rIL-35. Interestingly, while cyclin E and D1 levels are comparable in all

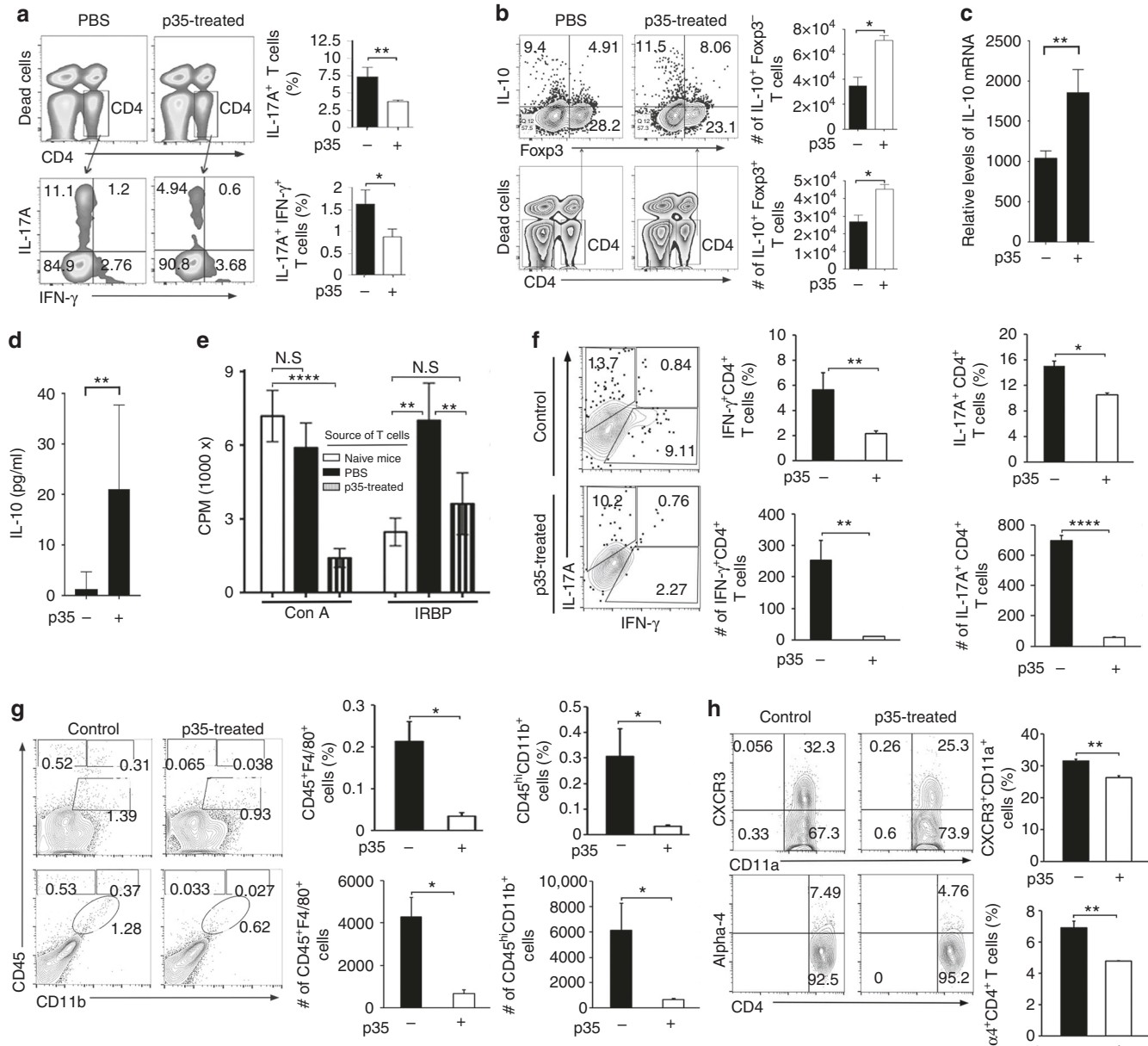

**Fig. 4** p35 inhibited the expansion of Th17 cells and reduced trafficking of inflammatory cells into the retina during EAU. **a**, **b** Intracellular cytokine analysis of IL-17-, IL-10-, Foxp3- or IFN-γ-expressing CD4[+] T cells in draining LNs on day 21 after induction of EAU. The cells were first stain with viability dye eFluor 450 (Invitrogen) to exclude dead cells and then subjected CD4 cell surface marker staining. The intracellular cytokine/protein staining finally was performed following cell permeabilization and cells were analyzed for IL-17, IFN-γ, IL-10, and/or Foxp3 expression **a**, **b**. Plots were gated on CD4[+] T cells and numbers in quadrants indicate percent of CD4[+] T cells expressing IL-10, IL-17, and/or IFN-γ. **c** cDNA was prepared from LN CD4[+] T cells and analyzed by RT-PCR. **d** Serum from untreated or p35-treated EAU mice were analyzed by ELISA. **e** Draining LN cells from untreated or p35-treated EAU mice were re-stimulated in vitro with Con A or IRBP for 3 days and assessed by Thymidine incorporation assay. **f**–**h** Retinae of mice that were either untreated or treated with p35 were isolated 21 days after induction of EAU, digested with collagenase and analyzed by FACS. *Graphs* indicate relative abundance of **f** IFN-γ[+] and IL-17[+] CD4[+] T cells; **g** CD45[+]CD11b[+] and/or CD45[+]F4/80[+] myeloid cells; **h** CXCR3[+]CD11a[+] or α4[+] CD4[+] T cells. Results represent at least three independent experiments and were analyzed using Student's t-test (two-tailed). Data are mean ± SEM. (*$P < 0.05$; **$P < 0.01$; ***$P < 0.001$; ****$P < 0.0001$)

samples, p35-treated B cells displayed higher level of p27[Kip1] compared to untreated cells (Fig. 1h), suggesting that IL-12p35 might inhibit B-lymphocyte proliferation by inducing cell-cycle arrest. Western blot analysis of TCR-activated CD4[+] T cells revealed that p35 could not activate STAT1, STAT3, or STAT4 but provided suggestive evidence that p35 might suppress lymphocyte proliferation by inhibiting IL-6-induced STAT3 activation (Fig. 1i) or IL-12-induced activation of STAT4 (Fig. 1j).

Taken together, these results suggest that IL-12p35 possesses intrinsic anti-proliferative activities.

It was however of interest to examine whether monomers and homo-dimers of IL-12p35 and/or Ebi3 can be detected in vivo during inflammatory immune responses of mice to infection, as may occur during sepsis. We therefore injected C57BL/6 mice with LPS. After 4 days, we isolated CD19[+] B cells from the spleen, lysed the cells and subjected the whole cell lysates to western blot analysis.

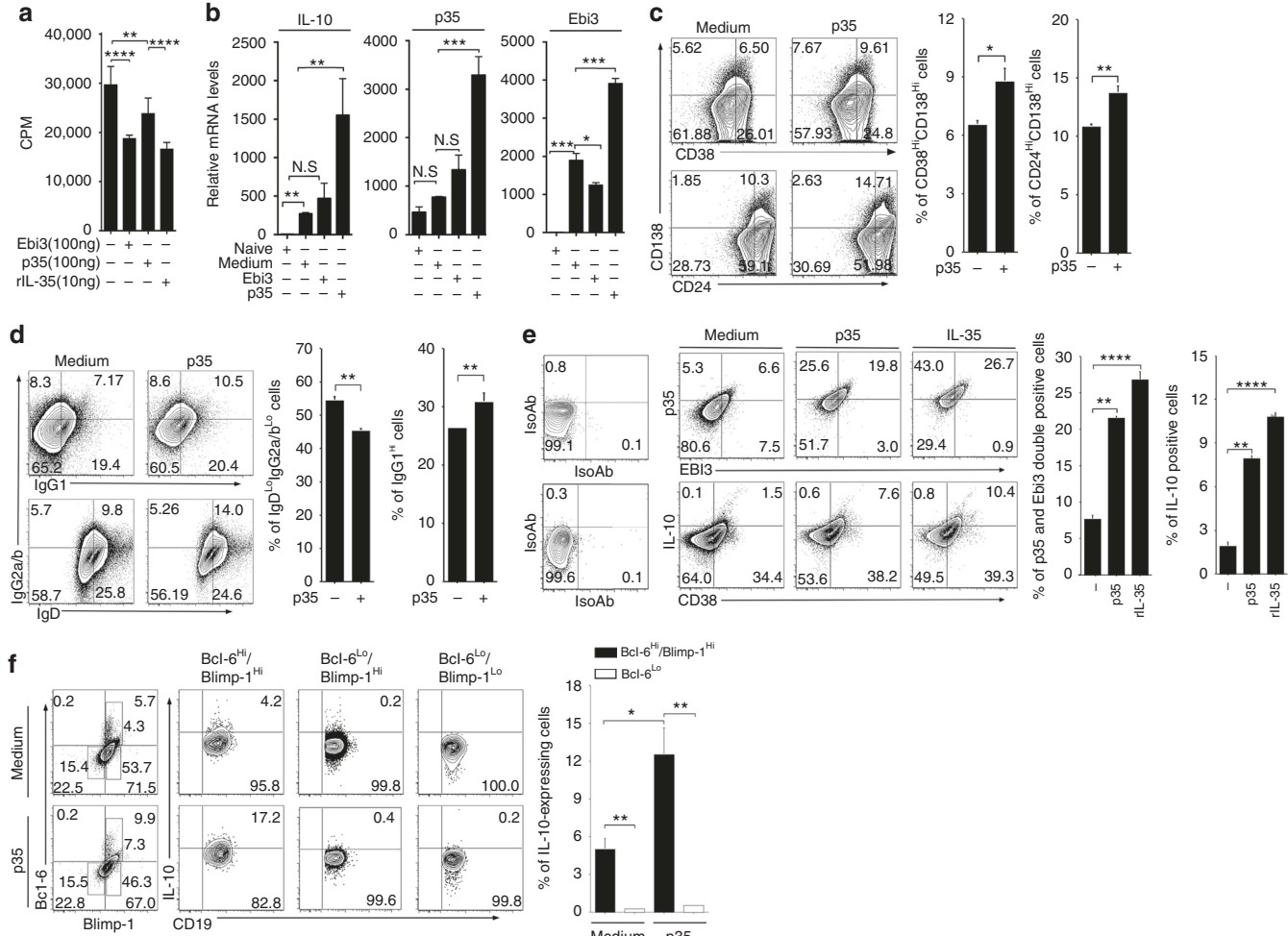

**Fig. 5** p35 induced expansion of IL-10- and IL-35-expressing B cells. **a** Primary mouse CD4[+] T cells were stimulated for 3 days with anti-CD3/anti-CD28 in medium containing rEbi3, p35, or rIL-35 and proliferative capacity of the cells was assessed by [³H]-thymidine incorporation assay. **b** CD19[+] B cells were activated with LPS in the absence or presence of p35 or rEBi3 and analyzed by qRT-PCR. **c–f** Purified primary mouse CD19[+] B cells were activated with LPS in the absence or presence of p35 and analyzed by FACS. The numbers in the *quadrants* indicate the percentages of IgG[+], IgD[+], CD138[+], CD38[+], and/or CD24[+] B cells. **d–f** CD19[+] B cells were activated with LPS in the absence or presence of p35 or rIL-35 and analyzed by the intracellular cytokine-staining assay for detection of B cells expressing IgG1, IgG2a/b, IL-12p35, Ebi3, Bcl-6, and Blimp-1 as indicated on the figures. Results represent at least three independent experiments and were analyzed using Student's *t*-test (two-tailed). Data are mean ± SEM (*P < 0.05; **P < 0.01; ***P < 0.001; ****P < 0.0001)

To our surprise, we detected not only the monomeric proteins but also, to a lesser amount, the p35–p35 and Ebi3–Ebi3 homodimers in the spleen cells of mice treated with LPS (Fig. 2a), suggesting that formation of p35–p35 homodimer may occur under conditions of intense inflammation. To confirm this finding, we next examined whether the p35–p35 homodimer also exists in vivo during experimental uveitis, an inflammatory disease of the eye. Analysis of whole cell lysate of the spleen by western blotting (under non-reduced condition) revealed significant expression of the p35 monomer in EAU mice treated with p35 compared to control mice (Fig. 2b). In contrast, we could not detect the p35–p35 homodimer (Fig. 2b), suggesting that significant amounts of the homodimer may not be produced in the periphery to allow its detection in the spleen during this localized inflammation of the immune privileged neuro-retinal tissue. It is also of note that Ebi3 is constitutively expressed with very little IL-12p35. The western blot analysis showing significant upregulation of p35 (Fig. 2b; left-most panel) thus provide suggestive evidence that the induced p35 couples with constitutively produced Ebi3 to produce IL-35 in p35-treated mice during intraocular inflammation.

**IL-12p35 suppresses autoimmune uveitis**. The function of IL-12p35 in vivo is complicated by the shared usage of IL-12p35 by IL-12 and IL-35. Moreover, the role of IL-12p35 in autoimmune disease remains unresolved and controversial as IL-12p35-deficient mice are protected against collagen-induced arthritis[29] while they develop exacerbated experimental autoimmune encephalitis (EAE)[30]. EAU shares essential immunopathogenic features with EAE and serves as an animal model of human uveitis. To directly examine the immunoregulatory functions of IL-12p35 during an organ-specific autoimmune disease, we induced EAU in C57BL/6J mice by active immunization with the ocular autoantigen, IRBP (interphotoreceptor retinoid-binding protein), and administered p35 or p35–p35 to some of the mice concurrently with immunization. Disease progression and severity were monitored by funduscopy, histology, optical coherence tomography (OCT) and electroretinogram (ERG). Consistent with published reports[31–33] funduscopic images taken 14 or 21 days after EAU induction showed that the untreated mice developed EAU characterized by optic disc inflammation (black arrow), vasculitis with cuffing (blue arrow), sclerotic blood vessels (white

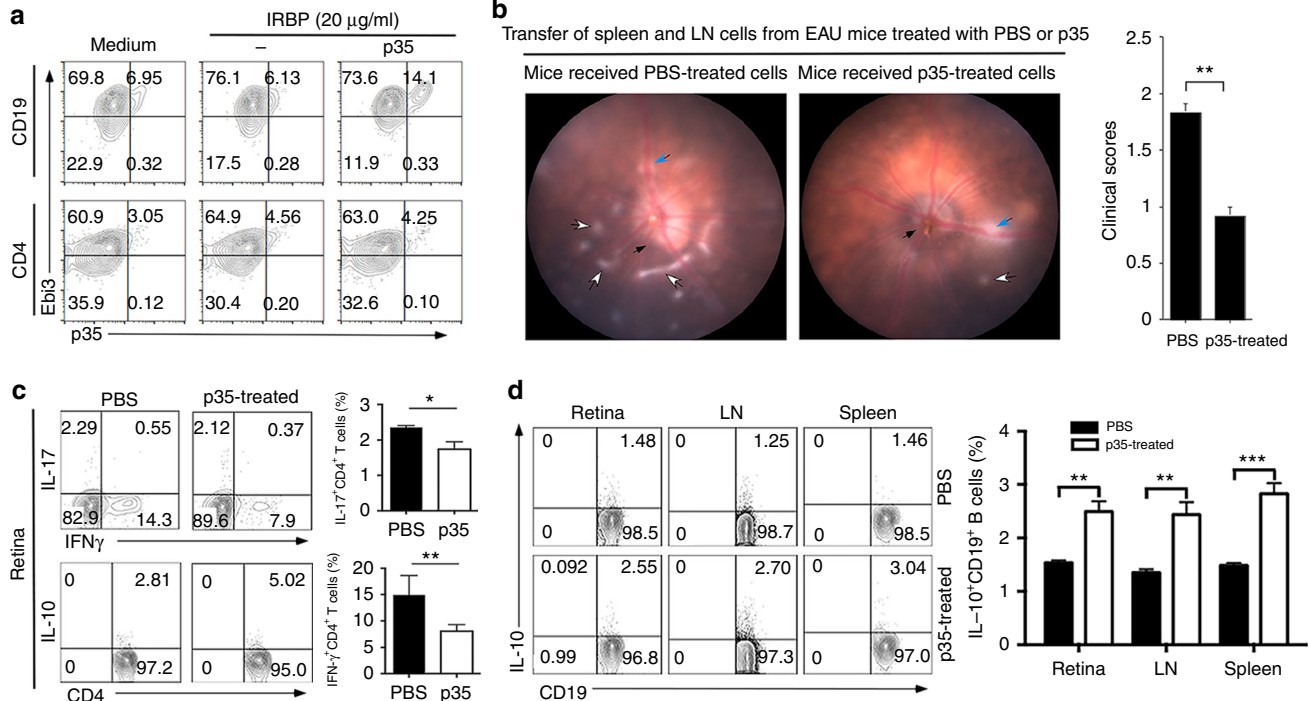

**Fig. 6** Uveitogenic T cells exposed to p35 in vivo loss capacity to transfer EAU. **a** Cells from the draining LN and spleens of EAU mice were re-stimulated ex vivo with IRBP in presence or absence of p35 and analyzed by intracellular cytokine assay. Numbers in the quadrants indicate the percentages of B or T cells expressing IL-35 (p35 and Ebi3). **b** EAU was induced in C57BL/6J mice by active immunization with IRBP and some of the mice were treated with p35 or PBS. Draining LN and spleen cells isolated 21 days post-immunization were re-stimulated ex vivo with IRBP and transferred ($1 \times 10^7$ cells) into naive syngeneic mice. Ten days after adoptive transfer the eyes were examined by funduscopy. Images reveal inflammation with blurred optic disc margins and enlarged juxtapapillary area (*black arrows*); retinal vasculitis with moderate cuffing (*blue arrows*); *yellow–whitish* retinal folding and infiltrates (*white arrows*). **c**, **d** Intracellular cytokine analysis of IL-17- or IFN-γ or IL-10-expressing CD4+ T cells in the retina **c** or IL-10-producing Breg cells in the retina, LN or spleen **b** on day 10 after adoptive transfer. Plots were gated on CD4+ T cells or CD19+ B cells and numbers in quadrants indicate percent of cells expressing IL-10, IL-17, and/or IFN-γ

arrow), and choroidal and retinal deposits (Fig. 3a, b). Histological analysis of the eyes 21 days after EAU induction further shows that the untreated mice developed several retinal folds (yellow asterisks) and increased numbers of inflammatory cells in the retina/vitreous (blue arrows). Evidence of these hallmark features of uveitis was not observed in mice treated with p35−p35 or p35 (Fig. 3c). The protective effects of p35 or p35−p35 was further confirmed by OCT analysis showing substantial presence of inflammatory cells in the optic nerve head and development of papillitis in untreated but not p35 or p35−p35-treated mice (Fig. 3d). Changes in ERG are indicative of alterations in neural activity and visual functions of photoreceptor and second-order neurons during ocular inflammation[34]. Photopic (light adapted) ERG responses on day 21 post-immunization were characterized by significantly lower b-wave amplitudes in eyes of untreated mice (Fig. 3e), suggesting that p35−p35 treatment might preserve cone signaling functions. Similar to the light-adapted responses, untreated mice elicited scotopic (dark adapted) ERG responses of significantly lower b-wave amplitudes, confirming the neuroprotective effect of IL-12p35 during EAU. It is important to note that we observed no substantial difference between p35 and p35−35 in these studies, suggesting that they can be used interchangeably.

**IL-12p35 inhibits Th17 and induces Treg cells during uveitis.** Th17 and IL-17/IFN-γ-expressing (DP-Th17) T cells increase in the spleen, draining lymph nodes (LN) and retina of mice with EAU and have been implicated in human and mouse uveitis[13, 25, 35]. It is of note that expansion of DP-Th17 cells is also associated with EAE[36] and Crohn's disease[37] and considered as a hallmark of uveitis[35]. We therefore isolated cells from the spleen, draining LN

and retina of the mice 21 days after induction of EAU and examined whether p35 reduced EAU severity by inhibiting the expansion of these pathogenic T-helper subsets. In line with our prediction, frequency of Th17 and to a lesser extent DP-Th17 cells was reduced in the spleen and draining LN of p35-treated mice compared to untreated mice (Fig. 4a). Reduction in the frequency of Th17-DP or Th17 cells in the treated mice suggests that p35 inhibited uveitis, in part, by targeting these pathogenic T-helper subsets. We further show that the decrease in Th17 cells was accompanied by increase of IL-10-expressing Foxp3+ and Tr1 regulatory T cells in the LN (Fig. 4b). Given the centrality of IL-10 in the suppression of inflammatory responses in several human and experimental autoimmune diseases, we validated the FACS data showing expansion of IL-10-producing regulatory T cells by RT-PCR (Fig. 4c) and ELISA (Fig. 4d). Interestingly, CD4+ T cells from p35-treated mice did not proliferate efficiently in response to IRBP or Con A (Fig. 4e), indicating that p35 might induce a hypo-proliferative state in vivo. Microglia and other myeloid cells also contribute to EAE pathology by enhancing Th17 effector functions[38]. We therefore examined whether p35 treatment inhibited the recruitment of lymphoid and myeloid cell types into the retina during EAU. We observed significant reductions in percentages of Th1, Th17 (Fig. 4f) or myeloid (Fig. 4g) cells in the retinae of the p35-treated mice (21 days after EAU induction). This correlated with reduced expression of CXCR3 and α4 integrin (Fig. 4h), suggesting that p35 might also mitigate uveitis by suppressing trafficking of inflammatory cells into the retina during EAU.

**IL-12p35 induces expansion of Bregs.** IL-35 suppresses inflammation by inhibiting Th17 cells while inducing expansion

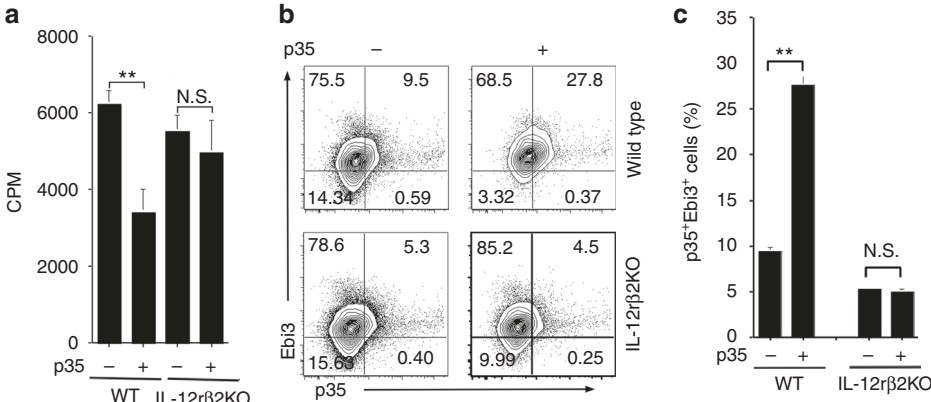

**Fig. 7** Immunosuppressive effects of p35 require IL-12Rβ2. **a** CD19[+] B cells from the spleen of WT or IL-12Rβ2KO mice were activated with LPS for 3 days in the presence or absence of p35 and proliferative capacity of the cells was assessed by [$^3$H]-thymidine incorporation assay. **b**, **c** CD19[+] B cells were activated with LPS in the absence or presence of p35 and analyzed by the intracellular cytokine-staining assay. The numbers in the quadrants indicate the percentages of IL-35-expressing B cells. Results represent at least three independent experiments and were analyzed using Student's $t$-test (two-tailed). Data are mean ± SEM (*$P < 0.05$; **$P < 0.01$; ***$P < 0.001$; ****$P < 0.0001$)

of IL-10 and/or IL-35-expressing T and B cells[21, 25, 39]. To investigate the relative contribution of IL-12p35 or Ebi3 to immune-suppressive functions of IL-35, we activated naïve CD4[+] T cells with anti-CD3 and anti-CD28 antibodies in medium containing p35, rEbi3 and/or rIL-35. While rEbi3 or p35 inhibited proliferation of the primary CD4[+] T cells, the growth inhibitory effect induced by either subunit alone was less compared to rIL-35 (Fig. 5a). While consistent with the data presented in Fig. 1f, it is interesting to note that effects of p35 and Ebi3 on T cell proliferation in Fig. 5a is less dramatic. This difference can be attributed to inherent variability of thymidine incorporation assays, especially when performed at different times and with different batches of Thymidine. In a previous report, we showed that IL-35 induces the expression of IL-10 and IL-35, as well as the expansion of IL-10- and IL-35-expressing B cells[25]. We therefore examined whether co-expression of IL-12p35 and Ebi3 is required for transcriptional activation of genes that code for IL-10 and/or IL-35. We activated CD19[+] B cells with LPS in medium containing p35 or rEbi3 and whereas p35 induced significant upregulation of IL-10, p35 and Ebi3 mRNAs, rEbi3 had modest or no effect on transcription of these genes (Fig. 5b). Taken together, these observations suggest that the inhibition of lymphocyte proliferation by IL-35 may derive from synergistic inhibitory effects of IL-12p35 and Ebi3, while transcriptional activation of *Il10*, *Il12a*, and *Ebi3* mainly requires IL-12p35. Thus, each IL-35 subunit may exert distinct and overlapping effects on lymphocytes that can be exploited therapeutically. Recent reports have also shown that IL-35 induces the expansion of IL-10-expressing and IL-35-expressing CD138[+] B cells[26, 40]. To examine effects of p35 on CD138[+] B cells, we activated CD19[+] B cells with LPS and cultured the cells in medium with or without p35. FACS and intracellular cytokine-staining analyses show that p35 induced expansion of IL-10-producing B cells characterized by the CD38[hi]Bcl6[hi]Blimp-1[hi] phenotype (Fig. 5c–f), suggesting that in addition to activation of anti-inflammatory genes, IL-12p35 can also induce the expansion of Breg cells and possibly promote the development of B cells toward plasma cell differentiation.

**Breg cells are induced in vivo by p35 treatment**. We have shown that p35 induces B cells in vitro and also inhibits autoimmune inflammation induced by active immunization of mice with IRBP. We next examined whether B cells induced in the spleen of EAU mice by p35 can be used to suppress uveitis. EAU was induced in

C57BL/6J mice by active immunization with IRBP and some of the mice were treated with PBS or p35. Draining LN and spleen cells were isolated on day 21 post-immunization and re-stimulated ex vivo with IRBP in medium containing PBS or p35. FACS and intracellular cytokine-staining analyses of CD19 [+]-gated cells or CD4[+]-gated cells indicate marked increase of IL-35-expressing CD19[+] cells in mice treated with p35 compared to PBS, reflecting enhanced expansion of IL-35-producing B cells in the spleen and LN of the p35-treated mice (Fig. 6a). Next, we adoptively transferred 10 million of the ex vivo re-stimulated cells from the spleen and LN of the EAU mice into naive syngeneic mice and assessed the development of EAU 10 days later. Whereas mice that received uveitogenic cells from the PBS-treated mice developed full-blown EAU 10 days after the adoptive transfer of the IRBP-specific pathogenic T cells, mice that received uveitogenic cells from the p35-treated mice had mild disease as indicated by funduscopy (Fig. 6b). This result suggests that in vivo exposure of the EAU-inducing uveitogenic T cells to p35, attenuated their capacity to transfer uveitis. Consistent with EAU induced by active immunization with IRBP, the percentage of Th1 and Th17 cells was decreased in the retinae of mice that received p35-treated uveitogenic cells and coincided with increase of IL-10-producing CD4[+] T cells (Fig. 6c). Similar analysis of cells from the retinae, lymph nodes and spleen revealed significant increases in IL-10-producing Breg cells in the EAU mice exposed to p35 in vivo compared to control mice treated with PBS (Fig. 6d), providing evidence that p35 might suppress EAU, in part, by inducing B cells in vivo.

**Immune-suppressive effects of IL-12p35 require IL-12Rβ2**. IL-35 mediates its effects in lymphocytes through signaling pathways that require the IL-12 family cytokine receptor, IL-12Rβ2[1, 25, 41]. Here, we have used mice that do not express IL-12Rβ2 to investigate whether the anti-inflammatory activities of p35 is mediated through the activation or suppression of signals downstream of the IL-12Rβ2 receptor. We isolated cells from the spleen of WT or IL-12Rβ2KO mice, sorted CD19[+] B cells and activated the cells with LPS for 3 days in the presence or absence of p35. In line with data presented above (Fig. 1h), p35 suppressed the proliferation of the WT B cells whereas the loss of IL-12Rβ2 in B cells abrogated the inhibitory effect of p35 (Fig. 7a). We next examined whether the loss of IL-12Rβ2 would also inhibit p35 induced expansion of regulatory B cells. Sorted CD19 [+] B cells from the spleen of WT or IL-12Rβ2KO mice were

activated with LPS for 3 days in the presence or absence of p35 and analyzed by the intracellular cytokine-staining assay. Consistent with data shown in Fig. 5e, stimulation of WT B cells with p35 induced expansion of IL-35-expressing B cells (Fig. 7b, c). In contrast, stimulation of IL-12Rβ2 deficient B cells with p35 did not induce expansion of the IL-35-producing B cells (Fig. 7b, c). Taken together, these observations suggest that p35 requires IL-12Rβ2 to mediate its anti-inflammatory activities and that it might do so by interfering with critical signaling pathways that are dependent on this important IL-12 family receptor subunit.

## Discussion

Our current understanding of the immunobiology of IL-35 or its subunits (IL-12p35 and Ebi3) is still rudimentary. However, recent reports indicate that IL-35 and IL-35-producing regulatory B (i35-Breg) and T cells (iTR35) are critical regulators of autoimmune diseases and cancer progression[1, 10, 25, 26]. In contrast to subunits of IL-12 and IL-23, which are secreted as covalently linked heterodimers, IL-12p35 and Ebi3 are secreted as independent subunits and published reports indicate that the two subunits associate during inflammatory conditions to form the bioactive heterodimer[1, 10, 42]. In this study, we have used recombinant IL-12p35 and Ebi3 to directly examine whether these IL-35 subunit proteins possess intrinsic immune-suppressive activities independent of their heterodimeric partner. We found subtle differences between the biological activities of the heterodimeric IL-35 cytokine and each of its constituent subunit proteins. While Ebi3 and IL-12p35 inhibited lymphocyte proliferation, Ebi3 consistently exhibited a higher inhibitory effect compared to IL-12p35. However, the heterodimeric IL-35 cytokine exhibited higher suppressive activity compared to either subunit protein. On the other hand, IL-12p35 upregulated expression of IL-10, p35 and Ebi3 mRNAs while Ebi3 had negligible effect on the expression of these transcripts. In addition, IL-12p35 induced the expansion of IL-10 B and i35-B cells, albeit to a lesser extent as the native IL-35. Thus, IL-12p35 more so than Ebi3 appears to manifest some of the immune-regulatory functions that had hitherto been attributed to IL-35.

An important unresolved issue relates to the bioavailability of the heterodimeric IL-35 in vivo and in particular, how stability of the heterodimer is maintained and regulated under physiological conditions. It has been suggested that independent secretion of IL-12p35 and Ebi3 might ensure exquisite regulation of the duration of the immune-suppressive action of IL-35, thereby avoiding inducing a permanent state of immune suppression that would otherwise ensue if IL-35 were to persist as a covalently bound IL-12p35: Ebi3 heterodimer. The IL-35 subunit proteins are not disulfide-linked and Alanine/Serine-scanning mutagenesis studies have revealed that mutations of critical residues associated with dimerization of p35 or Ebi3 to form IL-12 and/or IL-27 could not prevent generation of p35: Ebi3 heterodimer, suggesting that IL-35 has unique criteria for subunit pairing that is distinct from other IL-12 members[43]. In a previous study, we produced recombinant heterodimeric IL-35 cytokine in transfected insect cells and the purified heterodimeric cytokine isolated from the insect cells supernatants was used to effectively ameliorate uveitis in mice[25]. Interestingly, in this study we produced recombinant IL-12p35 in insect cells and found that in addition to the secreted IL-12p35 monomeric protein; substantial amounts of p35-p35 homodimer was also present. We also detected both p35 and the p35-35 homodimer in mice immunized with LPS, suggesting that the p35–p35 homodimer may be produced in vivo in response to acute systemic inflammation as may occur in septicemia. On the other hand, while the p35 monomeric protein was detected in the spleen of EAU mice, we could not detect the

p35–p35 homodimer, indicating that significant amounts of the homodimer may not be produced in the periphery to allow its detection in the spleen during the localized inflammation of the immune privileged neuro-retinal tissue. In view of the fact that IL-12p35 and Ebi3 contain 7 and 4 cysteine as well as 10 and 3 methionine residues[22], respectively, it is conceivable that these amino acids may render p35 and Ebi3 "sticky" and might contribute to propensity to form p35:p35 or Ebi3:Ebi3 homodimers. This raises the possibility that the increase of these homodimers during intense inflammation could contribute to immune-suppression in vivo. On the other hand, increase in p35–p35 homodimer may also provide a mechanism for limiting the bioavailability of the heterodimeric IL-35 in vivo. It is however of note that Il12a gene transcription is stringently regulated and it's in vivo level is relatively low. Thus, while our data suggest that p35 or p35–p35 homodimer can be used as Biologics to treat inflammatory disease, their efficacy may require high and non-physiological levels of the protein.

Another important issue relates to the promiscuous chain pairing/sharing exhibited by IL-12 family cytokines. Pairing of an αsubunit protein with IL-12p40 is suggested to promote pro-inflammatory responses while pairing to Ebi3 is generally associated with immune-suppression. Validity of these assumptions, if proven correct, will undoubtedly have implications in context of efforts to develop Biologics based on IL-12 family cytokines. However, the difficulty of predicting the immunological outcome of various combinations of α/βIL-12 subunit proteins is underscored by the recent discovery of IL-39, a novel pairing of Ebi3 and IL-23p19 that mediates pro-inflammatory responses in Lupus-like mice[44]. In another study, a genetically engineered novel IL-12-like cytokine composed of IL-12p40/IL-27p28 was found to exhibit immunosuppressive activities and effective in treating uveitis[45], further confounding our understanding of how the various IL-12 α and β subunits might influence the outcome of host immune responses. Production of large amounts of IL-12 single chain subunit proteins in vivo also raises the possibility that the single chain proteins may compete for pairing, altering the pattern or repertoire of IL-12 family cytokines secretion and thereby providing an additional mechanism for regulating the nature, quality and/or outcome of the immune response. The chain-sharing theme also extends to receptor use, as several IL-12 family cytokines utilize the same receptor chains[1]. In this study, we have shown that p35 alone, in the absence of its heterodimeric partner Ebi3, antagonizes signaling pathways utilized by pro-inflammatory cytokines like IL-6 and IL-12. We have also shown that p35 mediated inhibition of lymphocyte proliferation and expansion of B cells required signaling through the IL-12Rβ2. This is reminiscent of the antagonism of IL-12-driven responses by the competing of p40 homodimers for binding to IL-12Rβ1[46, 47]. These additional layers of complexity highlight the importance of not only studying the immune-regulatory activities of the heterodimeric cytokines but also the intrinsic activities of the subunit proteins as we have done in this study.

In summary, we have presented the novel findings that p35 can suppress lymphocyte proliferation, induce expansion of B cells and ameliorate uveitis by promoting expansion of Tregs/Bregs while antagonizing pathogenic Th17 responses. These findings suggest that IL-12p35 possesses immune-regulatory functions that had hitherto been attributed to IL-35. However, these results do not necessarily indicate that IL-35 and IL-12p35 exert identical and redundant immune-suppressive functions in vivo. In fact, while p35 possesses intrinsic lymphocyte growth inhibitory effects, induced expansion of Breg cells and suppressed EAU, its immunosuppressive effects were less compared to IL-35. Moreover, IL-35 activates STAT1, STAT3 and STAT4 signaling

pathways[25, 39, 41], while p35 antagonized IL-6 and IL-12 signaling pathways and could not activate STAT pathways. The inhibitory effect of p35 on lymphocyte proliferation was also less compared to Ebi3, suggesting that the full complement of the nuanced immune-regulatory functions of IL-35 might require both Ebi3 and IL-12p35. Thus, IL-12p35 might be promoting expansion of Bregs while Ebi3 synergizes with p35 in suppressing lymphocyte proliferation. Finally, the thrust of this study was to circumvent the labor-intensive efforts required for the production of large amounts of IL-35 for therapeutic use. Our demonstration that IL-12p35 can recapitulate some of the immunosuppressive activities of IL-35 is exciting and offers promise for the therapeutic use of p35 for in vivo and in vitro expansion of B and T cells. Data presented here also suggest that other single chain IL-12 family protein subunits should be explored as a potential new class of therapeutic cytokines.

## Methods

**Mice**. Wild type C57BL/6j and IL-12Rβ2KO mice on C57BL/6j genetic background were purchased from Jackson Laboratory. Mice were maintained and used in accordance with NEI/NIH Animal Care and Use Committee (ACUC) guidelines (ASP Protocol # EY000262-19 & EY000372-14) and the study protocol was approved by the ACUC Committee. Both male and female mice of 6–8 weeks old were used and the mice were randomized for all the studies described.

**Production and characterization of mouse rIL-12p35 and rEbi3**. Mouse recombinant *Il12a* and *Ebi3* cDNA constructs were generated by recombinant PCR (*Il12a* forward: 5′-CGCGGATCCATTGGCCAGGGTCATTCCAGT-3′, reverse: 5′-CCGCTCGAGGGCGGA GCTCAGATAG-3′; Ebi3 forward: 5′-CGCGGATCCAGAAACAGCTCTCGTGGCTCT-3′, Ebi3 reverse: 5′-TCCCCGCGGGGGCTTATGGGGTGCACTTTCTAC-3′). The IL-12p35 or Ebi3 cDNA was cloned into a 3.6 kb pMIB vector containing an amino-terminal melittin (HBM) secretion signal sequence and polyhistidine tags to facilitate isolation and characterization and expression of the constructs was driven by Baculovirus immediate-early promoters from the polyhedrosis virus (Catalog # V8030-01; Invitrogen, Carlsbad, CA). The expression construct was then transfected into insect high five cells and stable transfectants were identified by drug selection (Blasticidin S; 100 μg/ml). To ensure that the recombinant clone expressed bona fide IL-12p35 or Ebi3, we isolated the expression vector (HBM-p35-Flag-His or HMB-Ebi3-V5-His) and verified that no mutations were introduced during cloning or drug selection by DNA sequencing. The recombinant protein(s) secreted by the insect cells was sequentially purified by Ni-NTA Purification system (Invitrogen), size-exclusion Centricon filtration and two consecutive cycles of fast performance liquid chromatography (FPLC) on Supercryl-200 and Superose-6 columns. The rIL-12p35 and Ebi3 proteins were further characterized by SDS-PAGE, Western blot/immunoprecipitation and sedimentation equilibrium ultracentrifugation.

**Induction of EAU**. EAU was induced by active immunization with 150 μg bovine interphotoreceptor retinoid-binding protein (IRBP) and 300 μg human IRBP peptide, amino acid residues 1–20 (IRBP$_{1-20}$) in 0.2 ml emulsion 1:1 v/v with Complete Freund's adjuvant (CFA) containing *Mycobacterium tuberculosis* strain H37Ra (2.5 mg/ml). Mice also received *Bordetella pertussis* toxin (0.3 μg/mouse) concurrent with immunization. On the day of immunization and every other day until day 13 post-immunization, some mice were treated with either p35 or p35–p35 (100 ng/mouse). The disease progression and severity was established and monitored by fundoscopy, histology, optical coherence tomography (OCT) and electroretinography (ERG) as described previously[25, 34, 48]. Investigator was blinded to the group allocation during the EAU experiments and when assessing disease outcome or score. Eyes for histological evaluation were harvested 21 days post-immunization, fixed in 10% buffered formalin and serially sectioned in the vertical pupillary-optic nerve plane. All sections were stained with hematoxylin and eosin. For adoptive transfer of EAU, EAU was induced by active immunization with IRBP and draining LN and spleen cells isolated from mice treated with p35 or control untreated mice were re-stimulated ex vivo with IRBP. The cells ($1 \times 10^7$) were then transferred to naive syngeneic mice I.V and 10 days later development of EAU was examined by fundoscopy.

**Imaging mouse fundus**. Fundoscopic examinations were performed at day 14 and 21 after EAU induction using a modified Karl Storz veterinary otoendoscope coupled with a Nikon D90 digital camera, as previously described[49]. Briefly, following systemic administration of systemic anesthesia (intraperitoneal injection of ketamine (1.4 mg/mouse) and xylazine (0.12 mg/mouse)), the pupil was dilated by topical administration of 1% tropicamide ophthalmic solution (Alcon Inc, Fort Worth, Texas). To avoid a subjective bias, evaluation of the fundus photographs was conducted without knowledge of the mouse identity by a masked observer. At

least six images (2 posterior central retinal view, 4 peripheral retinal views) were taken from each eye by positioning the endoscope and viewing from superior, inferior, lateral, and medial fields and each individual lesion was identified, mapped and recorded. The clinical grading system for retinal inflammation was as previously established[31, 33].

**Imaging mouse retina by SD-OCT**. Spectral-domain optical coherence tomography (SD-OCT) is a non-invasive procedure that allows visualization of internal microstructure of various eye structures in living animals. An SD-OCT system with 1180 nm center wavelength broadband light source (Bioptigen, NC) was used for in vivo non-contact imaging of the eyes. Before OCT imaging was performed, each animal was anesthetized and the pupils dilated. The anesthetized mouse was immobilized using adjustable holder that could be rotated easily allowing for horizontal or vertical scan scanning. Each scan was performed at least twice, with realignment each time. The dimension of the scan (in depth and transverse extent) was adjusted until the optimal signal intensity and contrast was achieved.

**Electroretinogram recordings**. Before the Electroretinogram (ERG) recordings, mice were dark-adapted overnight, and experiments were performed under dim red illumination. Mice were anesthetized with a single intraperitoneal injection of ketamine (1.4 mg/mouse) and xylazine (0.12 mg/mouse) and pupils were dilated with Midrin P containing of 0.5% tropicamide and 0.5% phenylephrine hydrochloride (Santen Pharmaceutical Co., Osaka, Japan). ERG was recorded using an electroretinography console (Espion E2; Diagnosys LLC, Lowell, MA, USA) that generated and controlled the light stimulus. Dark-adapted ERG was recorded with single-flash delivered in a Ganzfeld dome with intensity of −4 to 1 log cd s/m$^2$ delivered in 6 steps. Light-adapted ERG was obtained with a 20 cd/m$^2$ background, and light stimuli started at 0.3–100 cd s/m$^2$ in 6 steps. Gonioscopic prism solution (Alcon Labs, Fort Worth, TX, USA) was used to provide good electrical contact and to maintain corneal moisture. A reference electrode (gold wire) was placed in the mouth, and a ground electrode (subcutaneous stainless-steel needle) was positioned at the base of the tail. Signals were differentially amplified and digitized at a rate of 1 kHz. Amplitudes of the major ERG components (a- and b-wave) were measured (Espion software; Diagnosys LLC) using automated and manual methods. Immediately after ERG recording, imaging of the fundus was performed as previously described above.

**Lymphocyte proliferation assay**. B cells were stimulated with 1 μg/ml LPS (Sigma L2654) while CD4$^+$ T cells were cultured in plate bound anti-CD3 Ab (10 μg/ml) and medium containing anti-CD28 Ab (1 μg/ml) Clones 145-2C11 and 37.51, respectively; BD Biosciences). B cells or T cells were propagated in presence or absence of p35, p35–p35, rEbi3 or rIL-35. For some co-culture experiments, LN and spleen cells were stimulated with IRBP in medium containing p35 or rIL-35. After 72 h, cultures were pulsed with $^3$H-thymidine (0.5 μCi/10 μl/well) as described[50]. Presented data are mean CPM ± SEM of responses of 5 replicate cultures.

**Detection of cytokine-expressing lymphocytes by FACS**. Primary B cells isolated from the spleen/LN (sorted for CD19$^+$ or B220$^+$) were stimulated with LPS (1 μg/ml). CD4$^+$ T cells (>98%) from the spleen and/or LN were activated in plate-bound anti-CD3 Abs (10 μg/ml) and soluble anti-CD28 Abs (1 μg/ml) as recommended by the manufacturer (BD Pharmingen, San Diego, CA, USA) and as previously described[25]. For intracellular cytokine detection, cells were re-stimulated for 5 h with PMA (Sigma, P8139) (20 ng/ml) and ionomycin (Sigma I0634) (1 μM). Golgi-stop was added in the last hour and intracellular cytokine staining was performed using BD Biosciences Cytofix/Cytoperm kit as recommended (BD Pharmingen, San Diego, CA, USA). FACS analysis was performed on a Becton-Dickinson FACSCalibur (BD Biosciences) using protein-specific monoclonal antibodies and corresponding isotype control Abs (PharMingen, San Diego, CA, USA) as previously described[13, 25]. FACS analysis was performed on samples stained with mAbs conjugated with fluorescent dyes and each experiment was color-compensated. Dead cells were stained with dead cell exclusion dye (Fixable Viability Dye eFluor 450; eBioscience) and live cells were subjected to side-scatter (SSC) and forward scatter (FSC) analysis. Quadrant gates were set using isotype controls with less than 0.2% background.

**Quantitative PCR analysis**. Total RNA was extracted from lymphocytes and retinal cells using the TRIzol reagent according to the procedures recommended by the manufacturer (Life Technologies, Gaithersburg, MD). All RNA samples were digested with RNAse-free DNAse 1 (Life Technologies) for 30 min, purified by phenol/chloroform extractions, and precipitated in 0.4 M LiCl. RNA (10 μg), a commercial synthesis system (SuperScript III Reverse Transcriptase; Life Technologies), and oligo(dT) were used for first-strand synthesis as previously described. First-strand synthesis containing each mRNA sample but without reverse transcriptase was performed to control for possible DNA contamination; failure to obtain real-time PCR (RT-PCR) products with any of the PCR amplimers confirmed the absence of contaminating DNA. All cDNA preparations used were suitable for PCR amplification on the basis of efficient amplification of a β-actin sequence. RT-PCR was performed on a fast RT-PCR system (ABI 7500) and PCR parameters were as recommended by the manufacturer (TaqMan Universal PCR

Kit; Applied Biosystems). Primers and probes were from Applied Biosystems: *Il12a* (ABI, Mm00434169_m1), *Ebi3* (ABI, Mm00469264_g), *Il10* (ABI, Mm00439616_m1), *gapdh* (ABI, Mm99999915_g1), *actin b* (Mm00469264_g1) (Supplementary Table 1). The mRNA expression levels were normalized to the levels of GAPDH housekeeping gene.

**Western blotting analysis**. Preparation of whole cell lysates and performance of western blot analysis were as described in ref. [51]. Cell extracts (20–40 μg/lane) were fractionated on 4–12% gradient SDS-PAGE, and antibodies used were: pSTAT3 (Cell Signaling Technology, Danvers Massachusettes); IL-12p35, Ebi3, Cyclin E, cyclin -D1, P27Kip1, pSTAT1, pSTAT3, pSTAT4, STAT4, and β-actin (Santa Cruz Biotechnology, Santa Cruz, CA, USA) (Supplementary Table 2). Pre-immune serum was used in parallel as controls and signals were detected with HRP-conjugated secondary F(ab′)₂ Ab (Zymed Laboratories) using the ECL-PLUS system (Amersham, Arlington Heights, IL, USA).

**Statistical analysis**. Statistical analysis was performed by Student's $t$-test (two-tailed) at 95% confidence level using Prism 7 software. EAU scores were analyzed by non-parametric Mann–Whiney U-test (two-tailed). Asterisks denote $P$-value (*$P < 0.05$, **$P < 0.01$, ***$P < 0.001$, ****$P < 0.0001$).

**Data availability**. The authors declare that the data supporting the findings of this study are available within the article and its Supplementary Information files, or are available from the authors upon request.

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

## Acknowledgements

We thank Dr. Haohua Qian and Yichao Li (Visual function core, NEI, NIH) for technical assistance with OCT; Phyllis Silver (NEI, NIH) for EAU scoring of the eyes; Rashid Mahdi. M.J.M. for technical assistance with western blot analyses and Rafael Villasmil (NEI FLOW Cytometry Core facility) for assistance with FACS analysis.

## Author contributions

I.M.D. purified and characterized rIL-12p35 and rEbi3, conducted most of the EAU experiments, prepared the figures and edited the manuscript. C.H. conducted most of the EAU experiments, prepared the figures and edited the manuscript. J.K.C. Performed adoptive transfer studies, analyzed adoptive transfer data and western blot analysis of IL-12 stimulated cells. C.-R.Y. assisted with EAU experiments, FACS analysis and preparation of the figures and also edited the manuscript. R.W. Generated the rIL-12p35 and rEbi3 cDNA constructs, partially purified and characterized the secreted proteins and assisted with editing the manuscript. M.J.M. assisted with EAU experiments, disease scoring and fundoscopy. P.T.W. performed equilibrium ultracentrifugation analysis. R.R.C. provided expertise in the description and analysis of EAU experiments and assisted with editing of the manuscript. C.E.E. conceived, designed and supervised the project and wrote the manuscript.

## Additional information

**Competing interests:** The authors declare no competing financial interests.

