## [Peer review file · Nature Communications]

Reviewers' comments:

Reviewer #1

Expert in Bregs

(Remarks to the Author):

The manuscript by Dambuza et al. describes the generation and use of p35 homodimers and Ebi3 homodimers as novel methods for immunosuppressing autoimmunity, specifically experimental autoimmune uveitis. While potentially true, it is most troubling that the authors' previous publication (Nat Med 2014, 20:633-641) indicates that IL-35, the p35:Ebi3 heterodimer, is directly responsible for suppressing autoimmunity. Does the current communication mean that the authors' previous publication was wrong or needs to be further modified again (see online corrections Nature Medicine 30 May 2014)? The authors cite this earlier paper throughout the current manuscript, so this basic conflict needs to be rigorously addressed. Moreover, all previous publications describing IL-35 or p35:Ebi3 heterodimers demonstrate that the two protein subunits are not disulfide linked, while the authors claimed in their previous study that the IL-35 protein subunits are disulfide linked. Why was the existence of p35:p35 and Ebi3:Ebi3 disulfide-linked homodimers not previously noticed in all of those studies, particularly in reference 10 that was so highly cited in the current manuscript? Are these heterodimers what the authors were actually observing in their previous publication, explaining why the authors were able to reduce the heterodimer to the p35 and Ebi3 subunits? Unfortunately, the authors do not address any of these conflicting observations, which are made even more confusing by the current paper.

Specific comments:

The two manuscripts cited as a reference for the statement that the majority of p35 occurs in homodimer form (page 3, first paragraph) do not overtly demonstrate nor enumerate any p35 homodimerization, so it is difficult to comprehend the basis for the authors' comments. At the risk of being repetitive, of most concern is that one of those two cited manuscripts, published in Nature Medicine, was prepared by the same group that submitted the manuscript currently under review. This is concerning, because the implication then is that, due to a lack of controls in the previous Nature Medicine manuscript, the anti-inflammatory effects previously observed that were ascribed to the impact of IL-35 (p35-Ebi3 heterodimer) are most likely due to the effects of p35 homodimer. This would make sense, because in the first figure of the Nature Medicine manuscript, the authors show that the derived recombinant p35-Ebi3 heterodimer can be reduced to the monomeric components of the heterodimer. This directly conflicts with many other publications, including the tenth reference cited in this paper, and available informatics resources, all of which indicate that p35 and Ebi3 are not linked through cysteine bonds and so do not have disulfide bonds holding them together that can then be reduced to break apart p35 and Ebi3 into monomeric units. Some explanation is needed to be able to sort through this.

A key issue concerning this manuscript is the lack of data supporting the anti-inflammatory effects of p35 homodimers. PBS or media alone are not adequate controls. The authors never demonstrate the specificity of the primary cell response to p35 homodimers, which could easily be done with cells from commercially available IL-12R β 2, gp130, or IL27R knock-out animals, or the use of control insect cell supernatant fluid lacking p35, so it is unclear whether the alterations in cell proliferation and cytokine production are due to a specific cytokine-dependent signaling event or non-specific activation and/or cell death. As the p35 material inhibits the function of all cells tested, how do the authors know that it is not just toxic? It is difficult to tell whether the authors actually used purified recombinant protein or just used supernatant fluid from the transfected insect cells for each of the experiments, which must be clarified. Additionally, the authors never investigate the natural prevalence of p35 homodimers in vivo. If p35 is as ubiquitously expressed as the authors indicate, it should be easy to isolate the homodimers from primary cells, an analysis that would substantially strengthen the significance of this manuscript.

The last major issue that permeates this study is the absence of adequate controls and

explanations of what exactly was done in most of the experiments. This is particularly true for Figures 3 and 4, most notably in the flow cytometry experiments.

Other issues:

Although it is likely to be an oversight, the authors must cite the paper that preceded their publication demonstrating that B cells express IL-35 (Shen, P., et al. IL-35-producing B cells are critical regulators of immunity during autoimmune and infectious diseases. *Nature* 2014, 507:366-370).

Figure 1c is not interpretable as no information is provided as to what this is in either the text or the figure legend.

Supplemental Figure 1 describing IL-35 production for the current paper was not available to this reviewer.

In Figure 4b, the middle panel is difficult to understand as a y axis of 40,000 should be mathematically impossible given how the experiment should have been carried out. How many PCR cycles were run? Given the importance of demonstrating that p35 is expressed by B cells, the authors should also show a Northern blot or use another method to actually demonstrate that this is real. Why does medium induce Ebi3 expression in the right panel while naïve B cells do not express these transcripts? No details were provided as to how or how long the cells were cultured.

The statement on page 6-7 that "p35 induced expansion of IL-10-producing CD38hiBcl6hiBlimphi plasma B cells phenotype (Fig. 4c, 4d)" is confusing. What exactly is meant by this sentence? Plasma cells are not known to expand (proliferate). Do the authors mean that plasma cells are producing IL-10 and the number of plasma cells is increasing? If this is the intent of the comment, they need to demonstrate that the CD38hiBcl6hiBlimphi cells are actually plasma cells. While plasma cells are generally found within the CD38hiBcl6hiBlimphi B cell subset, not all CD38hiBcl6hiBlimphi B cells are plasma cells. Thus, the authors must demonstrate that their IL-10+ cells are actually secreting antibody before considering them plasma cells. More importantly, this flow cytometry analysis is devoid of any control staining so how do the authors know where to set their gates and how do they know that the cells that they are calling p35+ are actually producing p35? As presented, it is impossible to place any confidence in these data.

The manuscript itself is not in an appropriate form for publication. The authors do not specify critical procedural or reagent information (e.g. source of flow cytometry information) in the methods section and have inconsistent nomenclature (e.g. IL-35 versus rIL-35) throughout the manuscript. The methods for presenting the data are also inconsistent throughout the manuscript (e.g. contour versus dot flow plots), which makes the study difficult to read. How many times were experiments repeated? How representative are the data shown of the authors results? Additionally, the manuscript would benefit profoundly from grammatical editing.

Reviewer #2

Expert in uveitis

(Remarks to the Author):

General comments

This is an interesting and intriguing paper by a group which has previously identified IL35 as an important immunoregulatory cytokine. They have followed up this work to show that in fact the IL-12p35 chain of IL35 is the definitive immunoregulatory component of IL35. In the main, the data are supportive of this concept although there are some specific issues which may need to be addressed (detailed below). The work in addition raises more questions than it answers. In particular, the mechanism of how the functional immunosuppressive effects of a molecule like IL-

IL-12p35 can be modified by variable combination with other chains is not explained? The authors have performed some initial examination of this issue by comparing the in vitro immunosuppressive effects of the IL-12p35 with its constituent chains. This group has previously shown that IL-12p40 is required for expression of EAU (not referenced) and further analysis of these molecular combinations for instance in the transcriptional activation of genes for T or Breg or Treg subsets may be informative eg how does IL-12p40 eliminate the effects of IL-12p35? Furthermore, the suggestion that IL-12p35 may be used as an immunosuppressive agent will require considerable characterisation of its pharmacokinetics.

Specific comments

The clinical images (Fig 2) could be improved. They appear to be out of focus. However they do show significant evidence of retinal vasculitis which is not correlated in the histological images shown (retinal folds and vitreous cell infiltrate only are shown). The overall degree of histological damage shown does not seem to represent "severe" EAU. However, the reduction in inflammation after IL-12p35 treatment is clear.

The percentage of TH17-DP cells is very low: may not be sufficient to label the uveitis as severe.

Reviewer #3

Expert in IL-35

(Remarks to the Author):

In this manuscript, Dambuza et al provide evidence that recombinant IL-12p35, which is a shared subunit of IL-12 and IL-35, either alone or formed as dimer, has a potent effect in the suppression of autoimmune inflammation in the eye. Recombinant P35 inhibits T cell proliferation, Th17/Th1 responses and induces B-reg cells that produce IL-35 and IL-10. Previously this group showed that recombinant IL-35 has the same function, now they have found that this function can be attributed to p35. This observation appears to be contradictory to what they have reported earlier in Nature Medicine. Nevertheless, the finding that p35 alone is an effective therapeutic for autoimmune disease is significant, given the fact that IL-35 is hard to produce.

Although the major observations are convincing and solid, some other data and interpretation are not flawless.

First, it appears that the most dramatic effect of P35 is the suppression of Th17 response. However, the mechanism of this effect is not clearly understood. It is very hard to explain this by p35 induction of B reg cells, since the authors provided no evidence that B-reg cells are induced in vivo. P35 also does not appear to induce signaling, but acts to block Stat3 activation induced by IL-6.

Second, although in vitro culture experiments clearly show p35 induces B-reg cells, the in vivo evidence is lacking. The only relevant experiment is Fig.4e. However, this is again an in vitro experiment, as P35 has to be added in the culture to show induction. Thus, there is no evidence that links p35 anti-inflammatory effect and B-reg induction.

Third, some data presented are not very convincing. In Fig.3a, I can barely see any IFN γ +IL17+ cells, to present this population and discuss it are pointless. In Fig.3b, I can barely see any difference of IL-10+ cells.

Overall, the authors convincingly show that recombinant p35 inhibits autoimmune inflammation in the eye and that p35 can induce B-regs in vitro. However, the mechanism of its action in vivo are not explained by induction of B-reg cells, since there is no evidence that B-reg cells are induced in vivo by p35 treatment.

Reviewer #1:

The manuscript by Dambuza et al. describes the generation and use of p35 homodimers and Ebi3 homodimers as novel methods for immunosuppressing autoimmunity, specifically experimental autoimmune uveitis. While potentially true, it is most troubling that the authors' previous publication (Nat Med 2014, 20:633-641) indicates that IL-35, the p35:Ebi3 heterodimer, is directly responsible for suppressing autoimmunity.

Response:

We thank reviewer for this comment that highlights the apparent confusion relating to how our current findings differ from our previous publication (Nat Med 2014, 20:633-641). The lack of clarity stems from the constraint imposed by the Nature Medicine Letter format that limits Text to 1,500 words (excluding the introductory paragraph). After the initial review, the Nature Medicine Senior Editor found the findings interesting but felt that it was more appropriate for Nature Communications and urged us to consider direct transfer of the manuscript to Nature Communication. Opting for direct transfer to Nature Communication did not afford opportunity for detailed discussion of our previous manuscript that might have allayed the concerns raised here. As discussed below and elaborated in the revised manuscript, we do not believe that our current work contradicts findings reported in our 2014 Nature Medicine Article. It is however important to begin our response by correcting Reviewer's assertion that "The manuscript by Dambuza et al. describes the generation and use of p35 homodimers and Ebi3 homodimers as novel methods for immunosuppressing autoimmunity, specifically experimental autoimmune uveitis". The generation and use of IL-35 heterodimer were described in the 2014 Nat. Med. paper. The expressed goal of this study was to investigate whether recombinant IL-12p35 and/or Ebi3 can be used as novel Biologics for suppressing autoimmunity, specifically experimental autoimmune uveitis.

Comment #1:

Does the current communication mean that the authors' previous publication was wrong or needs to be further modified again (see online corrections Nature Medicine 30 May 2014)? The authors cite this earlier paper throughout the current manuscript, so this basic conflict needs to be rigorously addressed.

Response:

We take exception to the Reviewer's suggestion that our previous publication (Nature Medicine 30 May 2014) was wrong or needs to be further modified again because of our current manuscript. It appears that there is a misunderstanding of the central message of these two manuscripts.

(i) The online modifications noted by reviewer were mainly corrections of typographical errors such as omitting the middle initial of a co-author (Monika Dolinska should have been Monika B. Dolinska); miss-spelling of the name of another co-author (Sergey should have been Sergeev); misspelling Mann-Whitney (not Mann-Whiney) in Figure legends (Figures 3 and 4) and lowercase/uppercase errors such as Il-10⁻ instead IL-10⁻. In fact, these errors were made by the Copy Editor in the Epub version and were corrected in the final published version.

(ii) In our 2014 Nature Medicine paper we genetically engineered the mouse IL-35 (p35 and Ebi3) in insect cells using a bicistronic vector. In Supplementary Figure 1 of the manuscript (Nat Med. 20(6):633-41), we presented FPLC purification of the secreted IL-35 protein on Supercryl S-200 columns. The data clearly showed that in addition to the heterodimeric IL-35 cytokine, p35 and Ebi3 monomers were also present. We therefore subjected the partially purified preparation to two additional sequential purifications on Seporose-6 column chromatography to obtain the highly purified heterodimeric IL-35 (p35/Ebi3) cytokine. We confirmed that the

purified IL-35 was indeed a p35:Ebi3 heterodimer by Western blot/immunoprecipitations and sedimentation equilibrium ultracentrifugation. It is the highly purified rIL-35 cytokine (not the insect cell supernatant or partially purified preparation) that we used to treat mice and suppress uveitis. Our current study should be put in context of the labor-intensive procedures and time-consuming efforts required to routinely produce large quantities of IL-35 for therapeutic use. We sought to circumvent these difficulties by examining whether the individual subunit proteins (p35 or Ebi3) also possess biological activities that can be exploited therapeutically. The current finding that the p35 and p35 homo-dimer possess some of the activities of the heterodimeric cytokine does not in any way contradict the previous finding that the IL-35 heterodimeric cytokine suppresses uveitis or inflammation *in vivo*.

Comment #2:

Moreover, all previous publications describing IL-35 or p35:Ebi3 heterodimers demonstrate that the two protein subunits are not disulfide linked, while the authors claimed in their previous study that the IL-35 protein subunits are disulfide linked.

Response:

We did not claim in our previous study that the IL-35 protein subunits are disulfide linked, even though IL-12p35 and Ebi3 contain 7 and 4 cysteine as well as 10 and 3 methionine residues, respectively [1]. We are in complete agreement with Reviewer that the two pro-inflammatory members of the family, IL-12 and IL-23 are disulfide-linked heterodimers whereas IL-27 and IL-35 lack disulfide linkage, pair poorly and secreted in much lower amounts (especially IL-35) [2-6]. In fact, linear and structure-based Alanine/Serine-scanning mutagenesis studies have revealed that mutations of critical residues associated with dimerization of p35 or Ebi3 to form IL-12 and/or IL-27 could not prevent generation of p35:Ebi3 heterodimer, suggesting that IL-35 has distinct criteria for subunit pairing from other IL-12 members [4]. It may well be that Ebi3 and p35 are sticky proteins and can form heterodimer, or some higher ordered multimers *in vivo* by unknown mechanism. On the other hand, dissociation of IL-35 into its monomers may involve multiple mechanisms including oxidative state, P_H , salt concentration, chemical reagents such SDS, chaotropic salt, and high temperature during protein analysis. However, we did not address the issue of whether the rIL-35 was disulfide linked or not because it was beyond the scope of that study. In fact, we underscored that the recombinant IL-35 (rIL-35) is a *non-covalently associated* heterodimer of p35 and Ebi3 by showing that it migrated as a ~67-kDa heterodimeric protein on native, non-denaturing gel whereas on denaturing SDS gel it migrated as 33-kDa monomeric protein (Fig. 1b of the 2014 Nat Med paper). This was further confirmed by sedimentation equilibrium ultracentrifugation under reduced and non-reduced conditions (Nat Med. 20:633-41; Supplementary Figure 1). Finally, in the last sentence of Discussion of the 2014 Nat. Med. manuscript we conclude: "Thus, factors that regulate the stability of the non-covalently linked IL-35 (p35 and Ebi3) heterodimer... suggest an additional layer of complexity that may underlie the physiological regulation of IL-35". We have clarified these points in the revised manuscript.

Comment #3:

Why was the existence of p35:p35 and Ebi3:Ebi3 disulfide-linked homodimers not previously noticed in all of those studies, particularly in reference 10 that was so highly cited in the current manuscript? Are these heterodimers what the authors were actually observing in their previous publication, explaining why the authors were able to reduce the heterodimer to the p35 and Ebi3 subunits?

Response:

1. While we are not in a position to explain data of other investigators, in our 2014 study we documented the presence of a single-chain Ebi3 or p35 protein migrating as a 33-kDa monomeric protein on denaturing SDS gels, whereas rIL-35 migrated as a ~67-kDa heterodimeric protein on native, non-denaturing gel (Fig. 1b). Clearly, some higher ordered multimers including homo-dimers were also present in the crude insect cell supernatant. However, our focus was on the heterodimeric IL-35 cytokine: characterization, analysis of the physical association or biological activities of these other proteins were beyond the scope of that study. The recombinant IL-35 cytokine described and used in all studies reported in our previous publication was a highly purified rIL-35. We provided evidence that it is indeed a heterodimer of p35 and Ebi3 by demonstrating the dual reactivity of p35- and Ebi3-specific monoclonal antibodies with the single band corresponding to the rIL-35 (Fig. 1d).

2. As noted above, the labor-intensive procedures and time required to routinely produce large quantities of IL-35 for clinical studies led us to investigate whether the individual subunits (p35 or Ebi3) might possess immune-suppressive activities. During the course of producing the recombinant proteins in our current study, we discovered that insect cells transfected with IL-12p35 or Ebi3 pMIB expression vector also secreted p35 or Ebi3 monomer and homo-dimer proteins. However, the homo-dimer reduced to monomer under denaturing condition of SDS PAGE analysis (Fig. 1). Nevertheless, whether the p35 protein is a monomer or non-covalently associates as a homo-dimer was not of importance to our goal of exploiting its immune-suppressive activity therapeutically. Moreover, we have not found discernible difference between p35 monomer and the p35-p35 homo-dimer in terms of their immunosuppressive activities: They possess equivalent immunosuppressive activities and the capacity to suppress uveitis (Fig. 3).

Comment #4:

Unfortunately, the authors do not address any of these conflicting observations, which are made even more confusing by the current paper.

Response:

While we appreciate Reviewer's interest in our previous manuscript and acknowledge that further discussion of some aspects of the Nature Medicine paper in context of our current work may be warranted, we again emphasize that the focus and goals of the two studies are distinct. The focus of our 2014 Nature Medicine paper was to characterize the IL-35 cytokine with respect to its cognate receptor and signaling mechanism in B cells and to investigate its immunosuppressive function in autoimmune Uveitis. In contrast, our current work investigates whether the individual subunits (p35 or Ebi3) possess immune-suppressive activities that can be exploited therapeutically as candidate Biologics.

- The finding in our 2014 Nature Medicine article that rIL-35 suppressed autoimmune uveitis in mice by inducing Breg and Treg cells is consistent with a 2014 report in Nature showing the IL-35 is a critical regulator of another CNS autoimmune disease, experimental autoimmune encephalomyelitis [7].
- In our current study we have shown that the p35 subunit also possesses immune-suppressive actions that can be exploited as a Biologic to treat uveitis.

Although infusion of the relatively large amounts of the p35 subunit was effective in ameliorating uveitis, considering its relatively low levels *in vivo*, we do not imply or propose that p35 is a more important immunosuppressant than IL-35. We therefore do not believe that our current paper contradicts any aspect of our previous publication.

Comment #5:

The two manuscripts cited as a reference for the statement that the majority of p35 occurs in

homodimer form (page 3, first paragraph) do not overtly demonstrate nor enumerate any p35 homodimerization, so it is difficult to comprehend the basis for the authors' comments. At the risk of being repetitive, of most concern is that one of those two cited manuscripts, published in Nature Medicine, was prepared by the same group that submitted the manuscript currently under review. This is concerning, because the implication then is that, due to a lack of controls in the previous Nature Medicine manuscript, the anti-inflammatory effects previously observed that were ascribed to the impact of IL-35 (p35-Ebi3 heterodimer) are most likely due to the effects of p35 homodimer. This would make sense, because in the first figure of the Nature Medicine manuscript, the authors show that the derived recombinant p35-Ebi3 heterodimer can be reduced to the monomeric components of the heterodimer. This directly conflicts with many other publications, including the tenth reference cited in this paper, and available informatics resources, all of which indicate that p35 and Ebi3 are not linked through cysteine bonds and so do not have disulfide bonds holding them together that can then be reduced to break apart p35 and Ebi3 into monomeric units. Some explanation is needed to be able to sort through this.

Response:

1. The Reviewer's comment is well taken. We have deleted the statement that the majority of p35 occurs in homo-dimer form. We thank reviewer for pointing out our inadvertently omission of "monomer" from the sentence noted by Reviewer. We meant to note that: "in addition to p35/Ebi3 heterodimer, the rIL-35 insect cell culture supernatant also contained substantial amounts of p35 and Ebi3 monomers, homo-dimers and higher order aggregates, suggesting that p35 monomer and homo-dimer might possess intrinsic activities". Please see the various proteins present in the partially purified IL-35 preparation (Suppl. Fig. 1; Nat Med. 2014 20(6): 633-41).

2. In response to Reviewer's concern that the anti-inflammatory effects ascribed to IL-35 (p35-Ebi3 heterodimer) in our previous Nature Medicine manuscript could have derived from effects of p35 homo-dimer, we note here the following aspects of the recombinant IL-35 production strategy that preclude this possibility.

(i) Careful design of the bicistronic vector ensured unambiguous isolation and purification of the Ebi3-p35 heterodimeric cytokine. The Poly-Histidine (6xHis) tag (HIS-Tag) used for purification was placed at the C-terminus of Ebi3 and was not attached to IL-12p35. Thus, HIS-tag-affinity chromatography allowed elution and isolation of only the HIS-tagged Ebi3-p35 heterodimer (IL-35) and any other secreted HIS-tagged Ebi3 proteins (e.g. HIS-Ebi3, HIS-Ebi3-Ebi3 homo-dimer or HIS-Ebi3-multimers). Note: p35 or p35-p35 will not be retained and isolated by the HIS-Tag affinity chromatography.

(ii) The affinity purified HIS-tagged Ebi3 proteins were then fractionated using 2 sequential Centricon filtration filters of 30kDa and 70kDa, allowing enrichment of HIS-tagged proteins ranging in size between 30kDa and 70kDa.

(iii) The partially purified proteins were subjected to further purification by FPLC on Sephacryl S-200 and Sepharose-6 columns. Fractions corresponding to the requisite size were collected and confirmed by PAGE.

(iv) Evidence that the HIS-Tagged rIL-35 is a heterodimer of p35 and Ebi3 was based on Western blot analysis showing dual reactivity of p35- and Ebi3-specific monoclonal antibodies.

(v) Finally, all studies described including the EAU treatment utilized the highly purified rIL-35. Moreover, the new data provided in the revised manuscript show that the p35 homo-dimer is not readily detectable in unimmunized mice or mice with EAU (Fig. 2b). We hope that these points should allay concerns that the anti-inflammatory effects observed in the Nature Medicine manuscript could be attributed to a p35 homo-dimer that is not readily detectable during EAU.

3. As noted above, various biochemical analyses have shown that IL-35 lacks disulfide linkage and that criteria for pairing of the p35 and Ebi3 subunits is unique and distinct from other IL-12

members [2-6]. It has been suggested that Ebi3 and p35 may be “sticky proteins” and that this may explain why they remain as heterodimer during non-denaturing native gel electrophoresis but readily dissociates to its monomers under the harsh denaturing conditions of SDS-PAGE [4]. Nonetheless, we did not address the issue of whether the rIL-35 was disulfide linked or not in our Nat Med paper because it was beyond the scope of that study and not relevant to therapeutic use of the purified IL-35.

4. We take exception to the Reviewer’s criticism that our peer-reviewed Nature Medicine manuscript lacked controls. The control for rIL-35 was the dialyzed, filtered supernatant from insect cells transfected with the empty vector (pMIB), as explicitly stated in the manuscript. The data showed that effect of pMIB is indistinguishable from PBS. None of the 4 reviewers who examined and accepted this manuscript raised any concerns about lack of appropriate controls.

5. In our current study, we also used the pMIB vector to produce the recombinant rIL-12p35 or rEbi3 protein in insect cells. The proteins were purified as described above. Because the p35 and p35-p35 proteins were highly purified by 4 cycles of FPLC, we consider PBS as the appropriate control. I like to point out that most published studies that examine effects of cytokines such as IL-12, IL-10 or IL-6 etc. use PBS as control.

Comment #6:

A key issue concerning this manuscript is the lack of data supporting the anti-inflammatory effects of p35 homodimers. PBS or media alone are not adequate controls. The authors never demonstrate the specificity of the primary cell response to p35 homodimers, which could easily be done with cells from commercially available IL-12R β 2, gp130, or IL27R knock-out animals, or the use of control insect cell supernatant fluid lacking p35, so it is unclear whether the alterations in cell proliferation and cytokine production are due to a specific cytokine-dependent signaling event or non-specific activation and/or cell death. As the p35 material inhibits the function of all cells tested, how do the authors know that it is not just toxic? It is difficult to tell whether the authors actually used purified recombinant protein or just used supernatant fluid from the transfected insect cells for each of the experiments, which must be clarified.

Response:

1. All the studies described in this manuscript utilized highly purified p35 or p35-p35 protein and not supernatant fluid from the transfected insect cells. As noted above the proteins were isolated on HIS-Tag affinity columns, followed by further purification by FPLC on Sephacryl S-200 and Sepharose-6 chromatography columns.

2. We have previously demonstrated (Nat Med. 2014 20(6): 633-41) that dialyzed supernatant fluid from insect cells transfected with the pMIB empty vector was equivalent in its effects to PBS. We therefore consider PBS to be the appropriate control for these highly purified proteins. This is in line with normal practice of the use of PBS as control for analysis of cytokine effects.

3. We respectfully differ from the Reviewer that this manuscript lacks data supporting the anti-inflammatory effects of p35 homo-dimer. We have shown by fundoscopy (Fig. 3a), Optical Coherence Tomography (Fig. 3d) and electroretinogram (Fig. 3e) that the p35 homo-dimer inhibited inflammation in mice with uveitis. We would like to emphasize that regardless of whether p35 protein is a monomer or non-covalently associates as a homo-dimer, this proof-of-principle study clearly establishes that p35 or p35-p35 can potentially be used as biologics to treat uveitis.

4. We thank Reviewer for the suggestion to verify the specificity of the primary cell response to p35. As suggested by Reviewer, we have used IL-12 β 2R-deficient cells to demonstrate the specificity of the primary cell response to p35. As shown in Fig. 7 of the revised manuscript, p35 inhibited proliferation of WT B cells (Fig. 7a) and induced the expansion IL-35-expressing WT

B cells (Fig. 7b, 7c). In contrast, p35 did not inhibit proliferation of IL-12R β 2KO B cells (Fig. 7a) and could not induce expansion IL-35-expressing cells in cultures containing IL-12R β 2KO B cells (Fig. 7b, 7c). Furthermore, the results showing that the proliferation of IL-12R β 2KO B cells was not affected or inhibited in cultures containing p35 indicate that p35 is not toxic. These results are consistent with published reports that have shown that while IL-35 suppresses expansion of inflammatory lymphocyte subsets, it promotes the expansion of regulatory lymphocyte populations [7, 8]. We have discussed these results in the revised manuscript (Please see page 11, lines 20-23; page 12, lines 1-13; Figure 7 and its Legend).

Comment #7:

It is difficult to tell whether the authors actually used purified recombinant protein or just used supernatant fluid from the transfected insect cells for each of the experiments, which must be clarified.

Response:

As noted in response to Comment #6, we used highly purified p35 or p35-p35 homo-dimer. We did not use supernatant fluid from the transfected insect cells in any of the experiments described.

Comment #8

Additionally, the authors never investigate the natural prevalence of p35 homo-dimers *in vivo*. If p35 is as ubiquitously expressed as the authors indicate, it should be easy to isolate the homo-dimers from primary cells, an analysis that would substantially strengthen the significance of this manuscript.

Response:

It is important to note that we did not state or imply that there is a “natural p35 homo-dimer” that is prevalent *in vivo*. We merely produced a recombinant IL-12p35, purified a p35-homodimer present in the insect cell supernatant and evaluated whether it can be used as a Biologic to treat uveitis. Nonetheless, in response to Reviewer’s comment we have performed the experiment recommended by Reviewer and the results are described in Fig. 2 of the revised manuscript.

1. To directly address Reviewer’s question relating to the natural prevalence of p35 homo-dimers *in vivo*, we performed two experiments:

(i) We examined whether formation of p35-p35 homo-dimer can be induced during an immune response to infection, as may occur during sepsis, as simulated by injection of LPS. Western blot analysis of spleen cells extracts detected expression of p35 and p35-p35 homo-dimer (Fig 2a).

(ii) We next examined whether the p35-p35 homo-dimer exists *in vivo* during experimental uveitis. Again, Western blot analysis of spleen extracts revealed expression of the p35 monomer in EAU mice treated with p35 compared to control mice (Fig. 2b). In the same mice, however, we could not detect the p35-p35 homo-dimer (Fig. 2b), suggesting that significant amounts of the homo-dimer may not be produced in the periphery to allow its detection in the spleen during this localized inflammation of the immune privileged neuroretinal tissue.

2. The transcription of the *il12a* gene is under stringent regulation and its *in vivo* level is very low. Thus, we did not suggest or believe that p35 homo-dimers would be prevalent or physiologically relevant *in vivo*. However, we posited that large amounts p35 or p35-p35 homo-dimer could be used as Biologics to treat inflammatory diseases. These points are discussed in the revised manuscript. (Please see page 6, lines 13-23; page 7, lines 1-7; page 14, lines 1-17; Figure 2 and its Legend).

Comment #9

Although it is likely to be an oversight, the authors must cite the paper that preceded their

publication demonstrating that B cells express IL-35 (Shen, P., et al. IL-35-producing B cells are critical regulators of immunity during autoimmune and infectious diseases. *Nature* 2014, 507:366-370).

Response:

We thank reviewer for alerting us to this error. In fact, the second citation of the previous version of this manuscript was Shen et al. (*Nature* 2014, 507:366-370). Unfortunately an Endnote error caused substitution of that article by Shen P et al for a 2014 paper by Shen H et al. We have provided the correct citation for this important paper in the revised manuscript (see Ref #24).

Comment #10

Figure 1c is not interpretable as no information is provided as to what this is in either the text or the figure legend.

Response:

We thank the Reviewer for pointing this out. The figure merely indicated the Fraction numbers corresponding to the elution of the purified proteins on the Sepharose 6 chromatography column. We have deleted the figure, as it is sufficient to simply note this point in the text.

Comment #11

Supplemental Figure 1 describing IL-35 production for the current paper was not available to this reviewer.

Response:

The purification strategy for Ebi3 and its result are similar to Figure 1a and therefore probably redundant. We have therefore deleted the Figure and instead describe the result in the Text.

Comment #12

In Figure 4b, the middle panel is difficult to understand as a y axis of 40,000 should be mathematically impossible given how the experiment should have been carried out. How many PCR cycles were run?

Response:

The PCR was performed by the standard method recommended for Taqman PCR and the standard default condition for quantification by the Relative quantification method is 40 cycles. The value of the Y-axis is influenced by many factors including the relative efficiency of the different probe/primer and the thresh-hold cycle.

Comment #13

Given the importance of demonstrating that p35 is expressed by B cells, the authors should also show a Northern blot or use another method to actually demonstrate that this is real.

Response:

In response to reviewer comments we have provided Western blot evidence showing that B cells produce p35 (see page 6, lines 15-19; Figure 2a). In addition, we previously demonstrated by Chromatin immunoprecipitation (ChIP) assay, Immunoprecipitation/Western blotting assays and intracellular cytokine assay that activated B cells express p35 (Suppl. Figure 3[8]).

Comment #14

Why does medium induce Ebi3 expression in the right panel while naïve B cells do not express these transcripts? No details were provided as to how or how long the cells were cultured.

Response:

We thank Reviewer for bringing to our attention that we did not state how long the cells were

cultured in vitro. The cells were stimulated with LPS in RPMI medium for 3 days and this information has been added to the revised manuscript. Non-stimulated naïve B cells do not express Ebi3. The B cells in the other lanes were pre-activated with LPS, which induces low levels of Ebi3.

Comment #15

The statement on page 6-7 that "p35 induced expansion of IL-10-producing CD38^{hi}Bcl6^{hi}Blimphi plasma B cells phenotype (Fig. 4c, 4d)" is confusing. What exactly is meant by this sentence? Plasma cells are not known to expand (proliferate). Do the authors mean that plasma cells are producing IL-10 and the number of plasma cells is increasing? If this is the intent of the comment, they need to demonstrate that the CD38^{hi}Bcl6^{hi}Blimphi cells are actually plasma cells. While plasma cells are generally found within the CD38^{hi}Bcl6^{hi}Blimphi B cell subset, not all CD38^{hi}Bcl6^{hi}Blimphi B cells are plasma cells. Thus, the authors must demonstrate that their IL-10⁺ cells are actually secreting antibody before considering them plasma cells. More importantly, this flow cytometry analysis is devoid of any control staining so how do the authors know where to set their gates and how do they know that the cells that they are calling p35⁺ are actually producing p35? As presented, it is impossible to place any confidence in these data.

Response:

We agree with Reviewer that description of the data is unclear. Indeed, some IL-10-producing B cells are characterized by the CD38^{hi}Bcl6^{hi}Blimp^{hi} immunophenotype. However, there is significant debate as to whether they are plasmablasts or plasma cells. In response to Reviewer's comment, we have revised the statements to reflect the fact that the cells expanded in our cultures express these cell surface markers. All mention of plasma cells in this context is deleted in the revised manuscript. All the requisite controls were indeed performed but the strict word limitation of the original Nature Medicine Letter format precluded an in-depth description. The Nature Communication format has afforded us the opportunity for expanded description of our methodologies and we have done so in the revised manuscript.

Comment #16

The manuscript itself is not in an appropriate form for publication. The authors do not specify critical procedural or reagent information (e.g. source of flow cytometry information) in the methods section and have inconsistent nomenclature (e.g. IL-35 versus rIL-35) throughout the manuscript. The methods for presenting the data are also inconsistent throughout the manuscript (e.g. contour versus dot flow plots), which makes the study difficult to read. How many times were experiments repeated? How representative are the data shown of the authors' results? Additionally, the manuscript would benefit profoundly from grammatical editing.

Response:

Most experiments were performed at least 3 times and the number of times each experiment was performed is stated in the legend to each figure. Critical procedural or reagent information is provided in the revised manuscript. In terms of the nomenclature relating to IL-35 versus rIL-35, we have clarified that: (i) IL-35 denotes the native IL-35, while rIL-35 denotes the recombinant mouse IL-35 we have produced; rIL-12p35 is referred to when we discuss the subunit while p35 and p35-p35 homo-dimer denote the 2 moieties secreted in the insect cell cultures.

Reviewer #2:

General comments:

This is an interesting and intriguing paper by a group which has previously identified IL35 as an

important immunoregulatory cytokine. They have followed up this work to show that in fact the IL-12p35 chain of IL35 is the definitive immunoregulatory component of IL35. In the main, the data are supportive of this concept although there are some specific issues which may need to be addressed (detailed below).

Response:

We thank Reviewer for the supportive comment and for noting that our paper is interesting and intriguing.

Comment #1:

The work in addition raises more questions than it answers. In particular, the mechanism of how the functional immunosuppressive effects of a molecule like IL-12p35 can be modified by variable combination with other chains is not explained? The authors have performed some initial examination of this issue by comparing the in vitro immunosuppressive effects of the IL35 with its constituent chains. This group has previously shown that IL12p40 is required for expression of EAU (not referenced) and further analysis of these molecular combinations for instance in the transcriptional activation of genes for T or Bregs or Tregs subsets may be informative eg how does IL12p40 eliminate the effects of IL12p3? Furthermore, the suggestion that IL-12p35 may be used as an immunosuppressive agent will require considerable characterisation of its pharmacokinetics.

Response:

1. We thank reviewer for the astute comment relating to complexities of the immunobiology of IL-12 family cytokines; in particular how different combinations of the various IL-12 alpha and beta subunits might influence the outcome of host immune responses. As noted in the text, despite sharing the IL-12p35 subunit, IL-12 (IL-12p35/IL-12p40) promotes inflammatory responses while IL-35 (IL-12p35/Ebi3) induces regulatory responses and this led to the notion that pairing of α -subunit proteins with IL-12p40 is pro-inflammatory while pairing with Ebi3 is immunosuppressive. However, recent studies cited in the revised manuscript underscore the difficulty of predicting the immunological outcome of various combinations of alpha/beta single chains. For example, the new IL-12 family member, IL-39, is a novel pairing of Ebi3 and IL-23p19 and contrary to prediction, IL-39 mediates pro-inflammatory responses in Lupus-like mice [9]. On the other hand, a genetically engineered novel IL-12-like cytokine composed of IL-12p40 and IL-27p28 is immunosuppressive and was found to be effective in treating uveitis [10], confounding understanding of how the various IL-12 alpha and beta subunit pairing might influence the outcome of host immune responses. This complexity thus highlights the importance of not only studying the immune-regulatory activities of the heterodimeric cytokines but also the intrinsic activities of the subunit proteins as we have done in this study. In response to Reviewer comment we have expanded discussion on the biology of IL-12 subunit proteins and how promiscuous chain pairing of the alpha and beta IL-12 subunits can be exploited therapeutically. (Please see page 3, lines 22-23; page 4, lines 1-11; page 14, lines 18-23; page 15, lines 1-18).

2. Prompted by the Reviewer's comment, we have added citation of our previous paper showing that IL-12p40-deficient mice are resistant to EAU, suggesting that endogenous IL-12 and/or IL-23 is required for expression of EAU [11]. We have also provided additional citations of studies indicating that besides forming heterodimers with either IL-12p35 or IL-23p19, the IL-12p40 subunit is also secreted independently as a monomer or disulfide-linked homo-dimer with increased levels of IL-12p40 detected in serum of patients with pulmonary sarcoidosis or multiple sclerosis [12, 13]. Interestingly, the IL-12p40 homo-dimer binds the IL-12 receptor and antagonizes IL-12 activity and anti-IL-12p40 treatment has a protective effect on the neurological dysfunction in MS patients [14, 15]. It is therefore notable that new data provided in

revised manuscript show that the capacity of p35 to antagonize IL-12 signaling (Fig. 1j) requires IL-12R β 2 (Fig. 7), suggesting a plausible mechanism for immunosuppressive effects of p35.

3. We agree with Reviewer that our proof-of-concept study suggesting that IL-12p35 maybe used as an immunosuppressive agent requires further studies to fully establish efficacy of p35 as a Biologic that can be used to treat uveitis and possibly other autoimmune diseases. Although beyond the scope of the current study, we intend to characterize the pharmacokinetics of p35 in mouse models of multiple sclerosis (EAE) and uveitis (EAU) as prelude to its development as a Biologic for CNS autoimmune diseases.

Comment #2:

The clinical images (Fig 2) could be improved. They appear to be out of focus. However they do show significant evidence of retinal vasculitis which is not correlated in the histological images shown (retinal folds and vitreous cell infiltrate only are shown). The overall degree of histological damage shown does not seem to represent "severe" EAU. However, the reduction in inflammation after IL-12p35 treatment is clear.

Response:

In response to Reviewer's comment relating the clinical images, we have replaced the low-resolution PDF images with high-resolution images in TIFF format. We agree that overall degree of histological damage shown does not look as "severe" as the pathology seen in susceptible strains such as the B10.RIII mouse or the Lewis rat, but this level of pathology is what we routinely observe in EAU disease in the C57BL/6 mouse strain. Therefore, it is severe for the C57BL/6 strain. We have now also provided new fundus images from our adoptive transfer studies that underscore the protective effects of p35 in EAU. (Please see page 11, lines 3-18; Figure 6 and Legend).

Comment #3:

The percentage of TH17-DP cells is very low: may not be sufficient to label the uveitis as severe.

Response:

We agree with Reviewer that the percentage of TH17-DP cells is relatively low compared to Th17 cells. That said, severity of uveitis is quantitated by tissue damage, not only the phenotype of effector cells. Furthermore, we have previously shown that both Th17 and Th1 cells are elicited and are independently pathogenic in EAU [16]. It is however of note that expansion of DP-Th17 cells is also associated with EAE [17] and Crohn's disease [18] and similar low percentages of TH17-DP versus TH17 cells are routinely observed in human and mouse uveitis [8, 19, 20]. Despite the relatively low abundance of Th17-DP cells, we consider detection of these cells as pathognomonic features or biological marker of disease. This point has been further discussed in the revised manuscript. (Please see page 8, lines 17-23; page 9, lines 1-4).

Reviewer #3:

In this manuscript, Dambuza et al provide evidence that recombinant IL-12p35, which is a shared subunit of IL-12 and IL-35, either alone or formed as dimer, has a potent effect in the suppression of autoimmune inflammation in the eye. Recombinant P35 inhibits T cell proliferation, Th17/Th1 responses and induces B-reg cells that produce IL-35 and IL-10. Previously this group showed that recombinant IL-35 has the same function, now they have found that this function can be attributed to p35. This observation appears to be contradictory to what they have reported earlier in Nature Medicine. Nevertheless, the finding that p35 alone is an effective therapeutic for autoimmune disease is significant given the fact that IL-35 is hard to

produce.

Response:

We are gratified that the Reviewer notes that the finding that p35 alone is an effective therapeutic for autoimmune disease is significant, given the fact that the heterodimeric IL-35 is hard to produce. In fact, it was the labor-intensive procedures and time-consuming efforts required to routinely produce large quantities of IL-35 for therapeutic use that led us to investigate whether individual subunits (p35 or Ebi3) might possess immune-suppressive activities that can be exploited to treat uveitis. As to the biological activities of IL-35 and p35, we do not feel that there is a contradiction with the previous paper. Other IL-12 subunit proteins have been shown to function autonomously as monomers or homo-dimers. For example, IL-27p28 alone has been shown to antagonize IL-27 signaling while the IL-12p40 homo-dimer antagonizes IL-12-driven responses and confers protection from some neurological dysfunctions of multiple sclerosis [14, 21-24]. In this case, the p35 monomer and homo-dimer have partial activity of the full cytokine. However, the heterodimeric IL-35 cytokine is clearly more effective in suppressing lymphocyte proliferation (Fig. 5a) and inducing the expansion of Breg cells (Fig. 5e). We mainly view the importance of the p35 as a potential Biologic. Thus, while infusion of relatively large amounts of the p35 can serve as a Biologic to treat uveitis, its efficacy is less than IL-35. We therefore do not believe that our current paper contradicts our previous publication. These points have been further discussed in the revised manuscript.

Comment #1:

Although the major observations are convincing and solid, some other data and interpretation are not flawless. First, it appears that the most dramatic effect of P35 is the suppression of Th17 response. However, the mechanism of this effect is not clearly understood. It is very hard to explain this by p35 induction of B reg cells, since the authors provided no evidence that B-reg cells are induced *in vivo*. P35 also does not appear to induce signaling, but acts to block Stat3 activation induced by IL-6.

Response:

In response to reviewer's comment we have provided new data showing that p35 can induce Breg cells *in vivo* (see Fig. 6 of the revised manuscript). We adoptively transferred uveitogenic cells from control mice with EAU or EAU mice treated with p35 into naïve syngeneic mice. Mice that received cells from the p35-treated mice had attenuated disease (Fig. 6b), which correlated with significant increases in Breg cells in the retina, draining lymph nodes and spleen (Fig. 6c, 6d). The p35-treated mice also had increased percentage of IL-10-producing T cells and decrease in Th1 cells in the retina (Fig. 6c). Although we do not completely understand the mechanism of immune suppression by p35, the new *in vivo* data presented here suggest that induction of Breg and Treg cells and suppression of pro-inflammatory T-helper cells might be an important component of the immune-suppressive function of p35. In addition, we have provided new data (Fig. 1j), that together with Fig. 1h and 1i, indicate that p35 might antagonize signaling pathways activated by inflammatory cytokines such as IL-6 or IL-12 and also inhibit expression of genes that regulate lymphocyte proliferation. These observations are indeed consistent with published reports of mechanisms utilized by other single chain IL-12 subunits. For example, p28 alone, in the absence of its normal partner Ebi3, limits production of IL-17 by antagonizing IL-6 signaling [21] while p40 homodimer antagonizes IL-12-driven responses and ameliorates some symptoms of MS by competing for binding to IL-12R β 1 [22-24]. (Please see page 11, lines 3-18; page 6, lines 1-12; page 3, lines 22-23; page 4, lines 1-11).

Comment #2:

Second, although *in vitro* culture experiments clearly show p35 induces B-reg cells, the *in vivo* evidence is lacking. The only relevant experiment is Fig.4e. However, this is again an *in vitro* experiment, as P35 has to be added in the culture to show induction. Thus, there is no evidence that links p35 anti-inflammatory effect and B-reg induction.

Response:

The Reviewer's comment is well taken. We have provided new data showing that p35 induces Breg cells *in vivo*.

1. As discussed above, new data from our adoptive transfer experiments show that mice that received cells from the p35-treated mice had attenuated EAU (Fig. 6b), which correlated with significant increases in Breg cells in the retina, draining lymph nodes and spleen (Fig. 6c, 6d) as well as increased percentage of IL-10-producing T cells in the retina (Fig. 6c). (Please see page 11, lines 3-18).
2. New Western blot analysis data show induced production of IL-35 (p35/Ebi3) in the spleen of EAU mice treated with p35 compared to control mice (Fig. 2b). (Please see page 6, lines 19-23; page 7, lines 1-7).
3. In terms of data presented in Figure 4e (Fig. 6a in the revised manuscript), we suggest that evaluation of the data should also take into consideration that while Breg cells increase in number during autoimmune diseases such as EAU or EAE, their *in vivo* levels are still quite low compared to Th17, Th1 or Treg cells. Thus it is necessary to reactivate the cells *ex-vivo* in order to evaluate their relative abundance. As shown in Fig. 6a, the percentage of IL-35-producing B cells increased in mice treated with p35 and we consider the result as a reasonable readout of the relative abundance of Breg cells in lymphoid organs of the control and treated mice. (Please see page 10, lines 17-23; page 11, lines 1-3).

Comment #3:

Third, some data presented are not very convincing. In Fig.3a, I can barely see any IFN γ +IL17+ cells, to present this population and discuss it are pointless. In Fig.3b, I can barely see any difference of IL-10+ cells.

Response:

1. We agree with Reviewer that the percentage of TH17-DP cells is relatively low compared to Th17 cells. Others have consistently documented presence of similarly low levels of TH17-DP cells in mouse autoimmune disease models such as EAE and EAU and in human diseases [17, 18, 20]. In response to Reviewer's comment we have toned-down our comment relating to this T-helper cell population. However, we would like to keep the Th17-DP data because despite their relatively low abundance many in the field consider detection of the Th17-DP population as biological marker of disease.
2. With regards to the data presented as Fig. 3b in the previous version of this manuscript (now Fig. 4b), we consistently found increased frequency of IL-10-producing TR1 and Foxp3 cells in mice treated with p35. Nonetheless, in response to Reviewer's comment we have provided new data from adoptive transfer studies, which corroborate our finding that p35-induces increase in IL-10-producing CD4⁺ T cells (please see Fig. 6c).

Comment #4:

Overall, the authors convincingly show that recombinant p35 inhibits autoimmune inflammation in the eye and that p35 can induce B-regs *in vitro*. However, the mechanism of its action *in vivo* are not explained by induction of B-reg cells, since there is no evidence that B-reg cells are induced *in vivo* by p35 treatment.

Response:

1. We have added 2 new data sets that we hope provide suggestive evidence that Breg cells are indeed induced *in vivo* by p35 treatment.

(i) In adoptive transfer experiments shown in Fig. 6, analysis of the retina, LN or spleen revealed significant increase in Breg cells in tissues of mice that received p35-treated cells compared to PBS-treated cells (Fig. 6d).

(ii) Following the induction of EAU by active immunization with IRBP, we analyzed splenic B cells by Western blotting and finding significant increase of p35 and Ebi3 proteins in extracts of mice treated with p35 (Fig. 2b) consistent with *in vivo* induction of IL-35 (p35/Ebi3) by p35.

2. We agree with reviewer that understanding of how p35 mediates its immunosuppressive effects *in vivo* is important. However, we feel that the comprehensive mechanistic studies needed to fully understand the mechanisms of p35 action is beyond the scope of this proof-of-principle study. As noted above, the impetus for this study stemmed from: (i) desire to produce a highly purified recombinant IL-12p35 and Ebi3 proteins; (ii) Establish for the first time whether either protein possesses intrinsic immune-regulatory activities independent of its heterodimeric partner; (iii) Examine whether p35 can be used as a Biologic for the treatment of a CNS autoimmune disease such as uveitis. In terms of the mechanisms that might underlie the effects of p35, we have established that recombinant p35 inhibits autoimmune inflammation in the eye and ameliorates ocular pathology by inducing Bregs and Treg cells while suppressing the expansion of pathogenic Th17 cells. We have also presented data indicating that p35 inhibits expression of cell cycle genes that regulate lymphocyte proliferation. We further show in the new data presented as Fig. 7, that while p35 inhibited proliferation of WT B cells (Fig. 7a) and induce expansion IL-35-expressing WT B cells, p35 could not inhibit proliferation of IL-12R β 2KO B cells (Fig. 7a) and could not induce expansion IL-35-expressing cells in cultures containing IL-12R β 2KO B cells (Fig. 7b, 7c). These results provide direct evidence that immune-suppressive functions of p35 derive in part from antagonizing IL-12R β 2. Moreover, our data showing that p35 antagonizes signaling pathways activated by pro-inflammatory signaling pathways induced by IL-6 and IL-12 is consistent with published reports that p28 or p40 homo-dimer alone, limits production of IL-17 or suppresses IL-12-driven responses, respectively [21-24]. We believe that these observations provide in part, mechanistic explanation of how single chain IL-12 subunits might regulate immune responses independent of their heterodimeric partners. Clearly, additional mechanistic studies and characterization of the pharmacokinetics of p35 during EAU will be required to fully develop p35 as an immunosuppressive agent.

References:

1. Devergne O, Birkenbach M, Kieff E. Epstein-Barr virus-induced gene 3 and the p35 subunit of interleukin 12 form a novel heterodimeric hematopoietin. *Proc Natl Acad Sci U S A* 1997; 94:12041-6.
2. Vignali DA, Kuchroo VK. IL-12 family cytokines: immunological playmakers. *Nat Immunol*; 13:722-8.
3. Beyer BM, Ingram R, Ramanathan L, Reichert P, Le HV, Madison V, et al. Crystal structures of the pro-inflammatory cytokine interleukin-23 and its complex with a high-affinity neutralizing antibody. *Journal of molecular biology* 2008; 382:942-55.
4. Jones LL, Chaturvedi V, Uyttenhove C, Van Snick J, Vignali DA. Distinct subunit pairing criteria within the heterodimeric IL-12 cytokine family. *Mol Immunol* 2012; 51:234-44.
5. Jones LL, Vignali DA. Molecular interactions within the IL-6/IL-12 cytokine/receptor superfamily. *Immunol Res* 2011; 51:5-14.

6. Yoon C, Johnston SC, Tang J, Stahl M, Tobin JF, Somers WS. Charged residues dominate a unique interlocking topography in the heterodimeric cytokine interleukin-12. *EMBO J* 2000; 19:3530-41.
7. Shen P, Roch T, Lampropoulou V, O'Connor RA, Stervbo U, Hilgenberg E, et al. IL-35-producing B cells are critical regulators of immunity during autoimmune and infectious diseases. *Nature* 2014; 507:366-70.
8. Wang RX, Yu CR, Dambuza IM, Mahdi RM, Dolinska MB, Sergeev YV, et al. Interleukin-35 induces regulatory B cells that suppress autoimmune disease. *Nat Med* 2014; 20:633-41.
9. Wang X, Wei Y, Xiao H, Liu X, Zhang Y, Han G, et al. A novel IL-23p19/Ebi3 (IL-39) cytokine mediates inflammation in Lupus-like mice. *Eur J Immunol* 2016; 46:1343-50.
10. Wang RX, Yu CR, Mahdi RM, Egwuagu CE. Novel IL27p28/IL12p40 cytokine suppressed experimental autoimmune uveitis by inhibiting autoreactive Th1/Th17 cells and promoting expansion of regulatory T cells. *J Biol Chem* 2012; 287:36012-21.
11. Tarrant TK, Silver PB, Chan CC, Wiggert B, Caspi RR. Endogenous IL-12 is required for induction and expression of experimental autoimmune uveitis. *J Immunol* 1998; 161:122-7.
12. Heinzl FP, Hujer AM, Ahmed FN, Rerko RM. In vivo production and function of IL-12 p40 homodimers. *J Immunol* 1997; 158:4381-8.
13. Shigehara K, Shijubo N, Ohmichi M, Kamiguchi K, Takahashi R, Morita-Ichimura S, et al. Increased circulating interleukin-12 (IL-12) p40 in pulmonary sarcoidosis. *Clin Exp Immunol* 2003; 132:152-7.
14. Brahmachari S, Pahan K. Role of cytokine p40 family in multiple sclerosis. *Minerva medica* 2008; 99:105-18.
15. Brahmachari S, Pahan K. Suppression of regulatory T cells by IL-12p40 homodimer via nitric oxide. *J Immunol* 2009; 183:2045-58.
16. Luger D, Silver PB, Tang J, Cua D, Chen Z, Iwakura Y, et al. Either a Th17 or a Th1 effector response can drive autoimmunity: conditions of disease induction affect dominant effector category. *J Exp Med* 2008.
17. Chen Y, Langrish CL, McKenzie B, Joyce-Shaikh B, Stumhofer JS, McClanahan T, et al. Anti-IL-23 therapy inhibits multiple inflammatory pathways and ameliorates autoimmune encephalomyelitis. *J Clin Invest* 2006; 116:1317-26.
18. Annunziato F, Cosmi L, Santarlasci V, Maggi L, Liotta F, Mazzinghi B, et al. Phenotypic and functional features of human Th17 cells. *J Exp Med* 2007; 204:1849-61.
19. Amadi-Obi A, Yu CR, Liu X, Mahdi RM, Clarke GL, Nussenblatt RB, et al. T(H)17 cells contribute to uveitis and scleritis and are expanded by IL-2 and inhibited by IL-27/STAT1. *Nat Med* 2007; 13:711-8.
20. Liu X, Lee YS, Yu CR, Egwuagu CE. Loss of STAT3 in CD4+ T cells prevents development of experimental autoimmune diseases. *J Immunol* 2008; 180:6070-6.
21. Stumhofer JS, Tait ED, Quinn WJ, 3rd, Hosken N, Spudy B, Goenka R, et al. A role for IL-27p28 as an antagonist of gp130-mediated signaling. *Nat Immunol*; 11:1119-26.
22. Cooper AM, Khader SA. IL-12p40: an inherently agonistic cytokine. *Trends Immunol* 2007; 28:33-8.
23. Gillessen S, Carvajal D, Ling P, Podlaski FJ, Stremlo DL, Familletti PC, et al. Mouse interleukin-12 (IL-12) p40 homodimer: a potent IL-12 antagonist. *Eur J Immunol* 1995; 25:200-6.
24. Khader SA, Partida-Sanchez S, Bell G, Jelley-Gibbs DM, Swain S, Pearl JE, et al. Interleukin 12p40 is required for dendritic cell migration and T cell priming after Mycobacterium tuberculosis infection. *J Exp Med* 2006; 203:1805-15.

Reviewers' comments:

Reviewer #1 (Remarks to the Author):

While the reviewers appreciate the time that the authors have put into preparing and revising this manuscript, many of the original concerns with this paper remain. A key issue is the lack of data supporting the notion that the observed anti-inflammatory effects derive from p35 homodimers. The authors attempt to demonstrate the specificity of the effect to p35 homodimers using commercially available IL-12R β 2-deficient mice to assay B cell proliferation, but do not show that the T cell phenotype alterations nor the in vivo immunosuppression observed are p35-dependent. This could have easily been done by repeating experiments from Figures 3-6 with IL-12R β 2-deficient T cells or mice. Further, the appropriate control for p35 injection is not PBS but rather lysate that has been purified in the same way as the lysate containing p35 was collected using supernatant from cells that lack p35 overexpression, as the authors disclose that "trace amount of processed protein" (page 24 line 644) were present in the lysate. All of the observed in vitro and in vivo data could be due to activation from whatever residual protein is present in the transferred lysate, meaning that these data are uninterpretable without some specific control, either injection of p35-free lysate or injecting the p35-containing lysate into IL-12R β 2-deficient mice. Finally, as mice deficient in p35 or Ebi3 are readily available, the authors should demonstrate that the proposed autocrine production of p35 observed in the manuscript (Figures 2 and 6) are not due to injection of p35-containing lysate that was then taken up by cells. As monomeric or homodimer p35 was injected into mice, and p35 was measured, how can one be confident that this a measure of endogenously produced p35 and not simply residual p35 from the injections/stimulations? What does this experiment look like in p35-deficient mice? These mice are readily available from Jackson Laboratories and would clarify whether the observed p35 in IRBP-immunized mice was endogenously produced or residual. Further, regarding Figure 2, the outcome of this uninterpretable experiment in no way indicates that p35 couples with Ebi3 during inflammation, but rather that Ebi3 expression is unaffected by p35 administration. The authors would have to do an EMSA or some other assay to show that the isolated Ebi3 was coupled with p35. Additionally, the authors should evaluate p35 and Ebi3 protein expression in the draining lymph nodes and eye, as was done in other sections of the manuscript.

Importantly, there are critical deficiencies in the description of how the authors purified and analyzed the heterodimers and homodimers. Specifically, the authors fail to disclose how the lysates were generated; the reference provided on page 21 in the Methods section goes to a previously published manuscript (ref. 50, Egwuagu, 2002, Journal of Immunology) that references another manuscript that does not disclose how the lysate was generated (Egwuagu, 1997, Journal of Immunology). The problem with not disclosing this information is that if the authors generated the non-reduced lysate by boiling without SDS, as opposed to not subjecting the lysate to any treatment whatsoever prior to analysis, the recombinant protein referred to as rIL-35 is not in fact rIL-35, because boiling without SDS would still de-couple non-covalently linked protein subunits. If this is the case, the product referred to as rIL-35 is likely covalently-linked Ebi3 homodimer, as Ebi3 homodimers would be the same size as rIL-35 and have the histidine tag that would be pulled down through a nickel-dependent purification. These concerns lead to a much more central biological issue: if p35 and Ebi3 homodimers link through disulfide bonds (as described on pages 5-6 lines 14-15 of the present manuscript), but p35 and Ebi3 do not form the heterodimer through disulfide linkages, as indicated in the response to reviewers provided by the authors, why would one ever expect for p35 and Ebi3 to naturally pair? It is then more likely that the effects observed by putative rIL-35 are actually due to Ebi3 homodimer; this model would fit with data presented in Figure 5A, where Ebi3 homodimer was observed to have comparable effects on T cell proliferation as rIL-35.

The authors still do not specify critical procedural or reagent information (e.g. source of flow cytometry information, method of isolating whole cell lysate) in the methods section and have inconsistent nomenclature throughout the manuscript. Moreover, the methods for presenting data lack appropriate controls (e.g. isotype or negative flow controls). The size of p35 varies dramatically throughout the results section, based on the included ladder, appearing to be

anywhere from 22-40 kDa; it is unclear whether this is due to a poorly run gel or if the proteins are in fact migrating at significantly different sizes. Also, in reference to Figure 1 (Page 6, lines 119-122), it is difficult for the reader to determine whether this difference in p27Kip1 was real or just a protein loading difference. To be convinced of the difference, it would be helpful if the authors would quantify the intensities using a densitometer read-out, as was provided for Figure 1J.

Reviewer #2 (Remarks to the Author):

This is a revised paper. Criticisms raised by this reviewer have been adequately answered. In particular, Figures and data are significantly improved. No further questions.

Reviewer #3 (Remarks to the Author):

In this resubmitted manuscript, Dambuza et al provide evidence that recombinant IL-12p35, which is a shared subunit of IL-12 and IL-35, either alone or formed as dimer, has a potent effect in the suppression of autoimmune inflammation in the eye. Recombinant P35 inhibits T cell proliferation, Th17/Th1 responses and induces B-reg cells that produce IL-35 and IL-10. Previously this group showed that recombinant IL-35 has the same function, now they have found that this function can be partially attributed to p35. The finding that p35 alone is an effective therapeutic for autoimmune disease is significant given the fact that IL-35 is hard to produce.

The major concerns raised by this reviewer were the lack of in vivo evidence for P35 induction of Bregs and potential mechanisms of Th17 suppression by P35. They now have added 2 new data sets (adoptive transfer experiments presented in Fig.6, western blotting in Fig.2b) to show that Breg cells are indeed induced in vivo by p35 treatment. Additionally, they provided some suggestive evidences that P35 might inhibit Th17 responses through antagonize IL-12/IL-6 signaling. Thus, the manuscript is considered to be significantly improved.

Point-by-point response to Referee 1

Comment #1:

While the reviewers appreciate the time that the authors have put into preparing and revising this manuscript, many of the original concerns with this paper remain. A key issue is the lack of data supporting the notion that the observed anti-inflammatory effects derive from p35 homodimers. The authors attempt to demonstrate the specificity of the effect to p35 homodimers using commercially available IL-12R β 2-deficient mice to assay B cell proliferation, but do not show that the T cell phenotype alterations nor the *in vivo* immunosuppression observed are p35-dependent. This could have easily been done by repeating experiments from Figures 3-6 with IL-12R β 2-deficient T cells or mice.

Response:

1. We thank reviewer for acknowledging the time put and good-faith effort made in addressing all concerns raised by reviewers. In the revised manuscript (NCOMMS-16-02775A), we provided evidence showing that while the p35-p35 homo-dimer was detected *in vivo* under conditions of intense inflammation induced by LPS, it was not detected in mice with uveitis, suggesting p35-p35 homo-dimer may not be physiologically relevant, at least in context of a localized organ-specific autoimmune disease such as uveitis. Nonetheless, the recombinant rIL-12p35 secreted by the insect cells contained the p35 monomer and p35-p35 homo-dimer and either protein exhibited immune-regulatory activities that suppressed lymphocyte proliferation *in vitro* (Fig.1) and ameliorated autoimmune uveitis in mice when administered *in vivo* (Fig. 3a, b, d, e). It is however important to note we found no discernible difference in immune-regulatory activities between p35 monomer and p35-p35 homo-dimer in these studies. In the revised manuscript and response to reviewer, we have provided additional comments relating to the p35 and p35-p35 homo-dimer (see page 5, lines 17-19; page 6, lines 1-8) and explicitly stated that we do not believe that the p35-p35 homo-dimer is physiologically relevant but may be important therapeutically if administered as a biologic for treatment of autoimmune disease such as uveitis. We therefore respectfully disagree with reviewer that the observed anti-inflammatory effects of IL-12p35 in uveitis derive from the p35-p35 homo-dimer, as the homo-dimer is not detectable during EAU. In fact, our data suggested that the p35-p35 homo-dimer may not be physiologically relevant in uveitis and that the anti-inflammatory effects derived mainly from the p35 monomer.

2. In response to previous comments made by reviewer, we had followed reviewer's recommendation and used commercially available IL-12R β 2-deficient mice to demonstrate specificity of the IL-12p35 subunit. We cultured IL-12R β 2 deficient B cells in the presence or absence of p35 and found that p35 suppressed the proliferation of the WT B cells (Fig. 7a) while the loss of IL-12R β 2 in B cells abrogated the inhibitory effect of p35 (Fig. 7a). Furthermore, stimulation of WT B cells with p35 induced expansion of IL-35-expressing B cells (Fig. 7b, 7c), while stimulation of IL-12R β 2-deficient B cells with p35 did not induce increase in the percentage of the IL-35-producing Breg cells (Fig. 7b, 7c). These results clearly establish the specificity of the IL-12p35 subunit. The results also suggest that p35 requires IL-12R β 2 to mediate its anti-inflammatory activities and might interfere with pathways dependent on IL-12 receptor signaling. It is however important to emphasize that the main goal of this study was to assess whether the IL-12p35 protein (p35 or p35-p35 homo-dimer) possesses intrinsic anti-inflammatory activities. Data presented in Fig. 2b shows substantial increase in p35 during EAU while the p35-p35 homo-dimer was not detected (Fig. 2b). These observations underscore our contention that the p35-p35 homo-dimer may not be physiologically relevant in uveitis and that the anti-inflammatory effects derived mainly from the p35 monomer.

3. Finally, we disagree with reviewer that that our study lacks experimental data showing that therapeutic administration of p35 homo-dimer can exert anti-inflammatory effects *in vivo*. As shown in the revised manuscript (Fig. 3a, b, d, e) the p35 homo-dimer exhibits immune-regulatory effects *in vivo* by ameliorating autoimmune uveitis as indicated by fundoscopy (Fig. 3a), Optical coherence tomography (Fig. 3d) and ERG (fig. 3e). Although administration of large doses of p35 or p35-p35 can each be exploited as Biologics to treat uveitis, our studies reveal that the p35-p35 homo-dimer is not detectable *in vivo* during IRBP-induced uveitis in mice, indicating that the anti-inflammatory effects derived mainly from increase of the p35 monomer *in vivo*.

Comment #2:

Further, the appropriate control for p35 injection is not PBS but rather lysate that has been purified in the same way as the lysate containing p35 was collected using supernatant from cells that lack p35 overexpression, as the authors disclose that “trace amount of processed protein” (page 24 line 644) were present in the lysate. All of the observed *in vitro* and *in vivo* data could be due to activation from whatever residual protein is present in the transferred lysate, meaning that these data are uninterpretable without some specific control, either injection of p35-free lysate or injecting the p35-containing lysate into IL-12R β 2-deficient mice.

Response:

We like to reiterate that the control used in the study is reasonable and appropriate. As explicitly stated in the revised manuscript, we did not use lysates or extracts of transfected cells in studies described in this study. The recombinant p35 or p35-p35 protein was secreted by the insect cells. The insect cell supernatant was harvested, sequentially purified by affinity chromatography on Ni-NTA column, size-exclusion centricon filtration and further purified by 4 cycles of FPLC. It is of note that most studies on the physiological or therapeutic effects of cytokines such as IL-12, IL-6 etc., invariably use PBS as control as is the case in our studies. Importantly, in our 2014 Nature Medicine manuscript we showed that supernatants from insect cells transfected with the control the empty vector (pMIB) was comparable to PBS (Nat Med 20:633-41).

Comment #3:

Finally, as mice deficient in p35 or Ebi3 are readily available, the authors should demonstrate that the proposed autocrine production of p35 observed in the manuscript (Figures 2 and 6) are not due to injection of p35-containing lysate that was then taken up by cells. As monomeric or homodimer p35 was injected into mice, and p35 was measured, how can one be confident that this a measure of endogenously produced p35 and not simply residual p35 from the injections/stimulations? What does this experiment look like in p35-deficient mice? These mice are readily available from Jackson Laboratories and would clarify whether the observed p35 in IRBP-immunized mice was endogenously produced or residual.

Response:

Reviewer’s assertion that p35-containing lysate was injected is incorrect; the mice were injected with highly purified p35 isolated from insect cell supernatant. In response to reviewer’s suggestion that autocrine production of p35 observed in the manuscript (Figures 2 and 6) derived from the p35 injected in mice and then taken up by cells, we have used wild type C57BL/6, p35-deficient and Ebi3-deficient mice to address this concern, as suggested by reviewer. We injected LPS and p35 into the mice, harvested cells from the spleen 48 hours

later and analyzed the cells by Western blotting. Please, see new Western blot data provided (provided for Review purpose only) showing that p35 induced expression of p35 and p35-p35 in WT and Ebi3-deficient but not by the p35-deficient cells. These results thus show that the observed autocrine production of p35 did not derive from the p35 injected in mice and then taken up by cells. Moreover, the Reviewer's suggestion that injection of p35 or p35-containing lysate could have been taken up by endogenous cells in the retina or another cell type is not supported by empirical studies. In fact, preponderance of data from studies where cytokines such as IL-12, IL-6, IL-23, IL-27 or IL-35 have been injected in mice or humans do not support the notion that injected cytokines are taken up by cells as suggested by Reviewer. Rather, such studies indicate that these cytokines bind their cognate receptors and mediate their biological activities by activating requisite STAT pathways. In addition, Reviewer's comment does not take into consideration data presented in the revised manuscript showing that cells cultured in the presence of p35 induced transcription of not only of the p35 mRNA but also Ebi3 and IL-10 mRNA (Fig. 5b). Secondly, the mice described in Fig. 2 were each injected with 100ng p35 intravenously and the Western blot analysis was performed 4 days post injection. Given the fact that the injected p35 would be highly diluted and distribution to several tissues, it is highly unlikely that the injected p35 would persist *in vivo* in sufficient levels to be detected after 4 days. Regardless of whether the injected p35 is taken up by cells or endogenously produced in response to the injection, the data provided in this study unequivocally reveal suppression of ocular inflammation and amelioration of uveitis in mice that received p35. Indeed the take-home message of the manuscript is that p35 exhibits therapeutic effects and maybe an efficacious Biologic for treatment of uveitis.

Comment #4:

Further, regarding Figure 2, the outcome of this uninterpretable experiment in no way indicates that p35 couples with Ebi3 during inflammation, but rather that Ebi3 expression is unaffected by p35 administration. The authors would have to do an EMSA or some other assay to show that the isolated Ebi3 was coupled with p35.

Response:

1. Regarding Figure 2, the purpose of the experiment was to examine whether p35-p35 homo-dimer is induced during immune response *in vivo*. The results show that p35-p35 homo-dimer formation indeed occurs under conditions of intense systemic inflammation such as LPS-induced sepsis but not during a localized organ-specific autoimmune disease such as uveitis.
2. The issue of whether p35 associates with Ebi3 to form IL-35 during inflammation is a settled matter as indicated by several published reports. In fact, in our 2014 Nature Medicine paper, we demonstrated associations of p35 and Ebi3 to form IL-35 by use of immunoprecipitation and Western blot assays. We disagree with reviewer that EMSA is an appropriate assay to demonstrate such protein-protein associations. EMSA (Electrophoretic Mobility Shift Assay) is used to characterize binding of a transcription factor to DNA and neither p35 nor Ebi3 is a transcription factors and binding of either protein to DNA is not relevant in context of this study. On the other hand, reports in the literature indicate that B cells constitutively express Ebi3 and the significant up-regulation of IL-12p35 induced in mice following administration of the p35 protein (Fig. 2b; left-most panel), provides suggestive evidence for IL-35 production during EAU.

Comment #5:

Additionally, the authors should evaluate p35 and Ebi3 protein expression in the draining lymph nodes and eye, as was done in other sections of the manuscript.

Response:

We have provided data showing increased p35 and Ebi3 protein expression in the draining lymph nodes of EAU mice treated with p35 protein (Fig. 6a). It is also well known that the retina is an immune privileged tissue with active mechanisms that function to restrict infiltration of lymphocytes even during intraocular inflammation. It is therefore remarkable to have observed significant increases of IL-10 producing regulatory B and T cells in the retina of mice treated with p35 compared to untreated mice (Fig. 6c, d). While it would be interesting to evaluate p35 and/or Ebi3 protein expression in the retina, it is technically difficult. As shown, the percentage of IL-10 expressing cells in retina is less than 3% of the few infiltrated T and B cells; thus Ebi3/p35 positive cells in the retina would be few and difficult to detect.

Comment #6:

Importantly, there are critical deficiencies in the description of how the authors purified and analyzed the heterodimers and homodimers. Specifically, the authors fail to disclose how the lysates were generated; the reference provided on page 21 in the Methods section goes to a previously published manuscript (ref. 50, Egwuagu, 2002, Journal of Immunology) that references another manuscript that does not disclose how the lysate was generated (Egwuagu, 1997, Journal of Immunology). The problem with not disclosing this information is that if the authors generated the non-reduced lysate by boiling without SDS, as opposed to not subjecting the lysate to any treatment whatsoever prior to analysis, the recombinant protein referred to as rIL-35 is not in fact rIL-35, because boiling without SDS would still de-couple non-covalently linked protein subunits. If this is the case, the product referred to as rIL-35 is likely covalently-linked Ebi3 homodimer, as Ebi3 homodimers would be the same size as rIL-35 and have the histidine tag that would be pulled down through a nickel-dependent purification. These concerns lead to a much more central biological issue: if p35 and Ebi3 homodimers link through disulfide bonds (as described on pages 5-6 lines 14-15 of the present manuscript), but p35 and Ebi3 do not form the heterodimer through disulfide linkages, as indicated in the response to reviewers provided by the authors, why would one ever expect for p35 and Ebi3 to naturally pair? It is then more likely that the effects observed by putative rIL-35 are actually due to Ebi3 homodimer; this model would fit with data presented in Figure 5A, where Ebi3 homodimer was observed to have comparable effects on T cell proliferation as rIL-35.

Response:

There appears to be a fundamental misunderstanding relating to how the p35 and p35-p35 homo-dimer were produced and purified. We have therefore provided a schematic illustration of the procedures used to produce and purify the recombinant IL-12p35 (Reviewer Fig. 2). As indicated above neither the p35 nor p35-p35 homo-dimer was generated from insect cell lysate. The recombinant IL-12p35 or Ebi3 proteins were genetically engineered using the pMIB expression vector encoding the efficient Honeybee Mellitin secretion signal peptide and the secreted proteins were harvested from the insect cell supernatant. Although expression of the proteins is also detectable in the cell lysate, it was important to utilize the mature post-transcriptionally modified secreted protein. Thus the p35 and p35-p35 homo-dimer utilized in our studies were purified from the supernatants of the insect cells and not from lysates of the transfected insect cells. Presumably, the cell lysates alluded to by Review were those derived from spleen cells of EAU mice or mice injected with LPS and the method used to prepare the lysates is as described by Egwuagu, 1997 and 2002; Journal of Immunology. Nonetheless, mass determination of the purified p35 or p35-p35 was determined by sedimentation equilibrium. With regards to the rIL-35 described in our Nature Medicine paper, similar approach was used to purify and characterize the cytokine and identity of the rIL-35 was

confirmed by the dual reactivity of antibodies to p35- and Ebi3-specific monoclonal antibodies in Western blotting and co-immunoprecipitation analyses (Nat Med 20:633-41).

Comment #7:

The authors still do not specify critical procedural or reagent information (e.g. source of flow cytometry information, method of isolating whole cell lysate) in the methods section and have inconsistent nomenclature throughout the manuscript.

Response:

Critical procedural and reagent information are provided in the revised manuscript. In terms of the nomenclature relating to IL-35 versus rIL-35, we have clarified that: (i) IL-35 denotes the native IL-35, while rIL-35 denotes the recombinant mouse IL-35 we have produced; (ii) rIL-12p35 is referred to when we discuss the subunit while p35 and p35-p35 homo-dimer is used to describe the 2 moieties secreted in the insect cell cultures.

Comment #8:

Moreover, the methods for presenting data lack appropriate controls (e.g. isotype or negative flow controls). The size of p35 varies dramatically throughout the results section, based on the included ladder, appearing to be anywhere from 22-40 kDa; it is unclear whether this is due to a poorly run gel or if the proteins are in fact migrating at significantly different sizes.

Response:

The requisite controls employed in FACS and Western blot analysis are provided in the revised manuscript. Antibodies used were from BD, R&D or eBioscience and use of these antibodies was as suggested by manufacturers.

Reviewers' comments:

Reviewer #1 (Remarks to the Author):

No comments

Reviewer #4 (Remarks to the Author):

This paper addresses whether the IL-35 subunit IL-12p35 possesses a previously unappreciated immune-regulatory function separate to its role in the formation of the IL-35 heterodimer IL-12p35/Ebi3. Although this study is extremely novel as (to my knowledge) this is a previously unexplored area of research as well as well-written and clearly explained there are some inconsistencies in the paper that need to be addressed.

One inconsistency in the data is the interchangeability of the use of the p35 monomer and p35-p35 homodimer. Although the authors show that proliferation of CD4+ T cells anti-CD3/CD28 stimulated CD4+ T cells is similar following exposure to p35 monomer and p35-p35 homodimer they do not discount that these two forms have different effects on B cells (Fig. 1g only shows the effect of the monomer on B cells) or other effects in vivo. To address this the authors must show the same data in Fig. 3 for the p35 monomer and p35-p35 homodimer e.g. Day 21 is only shown for the fundus images for the p35-p35 treated mice and no ERG analysis is shown for p35-treated mice. In fact from the data that is shown it appears that p35 leads to a more significant suppression of retinal inflammation than p35-p35 in treated mice. Indeed, it seems likely that as a monomer that p35 may more readily couple with Ebi3 in vivo increasing the amount of IL-35. With this in mind, it has also been previously shown that changing one subunit (e.g. IL-12p35) of the IL-12 family can change the balance between other family members (Vasconcellos et al., JI, 2011) and the authors do not measure IL-12, IL-27 or IL-23 levels following in vivo administration with IL-12p35.

Minor comments:

1. In the first section of the paper the authors describe the production of recombinant mouse p35 and Ebi3. However in Fig. 1 there is only data that describes the size and conformation of p35 (monomer/homodimer) following analysis of the secreted peptides by SDS-page. Could the authors please explain why they have excluded the data concerning the size and conformation of recombinant Ebi3?
2. In reference to Fig. 1h the authors comment that 'p35 treated cells displayed higher level of p27kip1....' but in the figure they only show the data from CD4+ T cells. Please change the text or add the B cell data to the figure.
3. In Fig. 2 the authors show that splenocytes from LPS-injected mice express p35-p35 and Ebi3-Ebi3 homodimers but splenocytes from IRBP-immunized mice do not. The authors suggest that is due to the intense inflammatory conditions induced by LPS. Although I agree with the authors this is the most likely explanation, it could be due to the location of the cells draining the site of inflammation. With this in mind, it would be informative to also show data from cell lysates obtained from the LN draining the inflamed eye in Fig. 2b.
4. In Fig. 4a (Th1/Th17) and Fig. 4c (proliferation) the authors show T cell phenotype in the draining LN but in Fig. 4b (Foxp3) the authors showing T cell phenotype in the spleen. Please either show the data from the same organs, all the organs, or provide an explanation as to why different organs have been chosen.
5. Please show representative plots for Fig. 4d-f.
6. In Fig 1f the authors show that the stimulation of CD4+ T cells with CD3/CD28 in the presence of the 100ng/ml of p35 results in a four-fold decrease in proliferation. However, in Fig 5a this affect is dramatically less indeed the authors go on to comment in the text that Ebi3 is much more efficient at suppressing proliferation than p35. Although thymidine incorporation assays can be extremely variable this effect seems drastically altered between these two figures. As the authors use this data to conclude that Ebi3 has a different effect to p35 it is important for the authors to

comment on this. Although it could be a simple explanation that these CD4+ T cells are from immunized mice, this is not clear in the legend or the text. In future work, it would be extremely interesting to look at the in vivo effect of Ebi3 on the development of autoimmune disease.

7. Please show an isotype control for the gating of Ebi3/p35 double positive cells and IL-10 for the different B cell subsets in Fig. 5e.

8. In Fig. 6a a mixture of lymphocytes from the draining LN and spleen are restimulated to look at the production of P35+EBI3+ cells, could the authors please provide an explanation or show these data from these organs separately.

9. Please show the summary data for Fig. 6c.

10. Although most of the proliferative data (Fig 1.f comparing monomer/homodimer, Fig. 5a comparing Ebi3, p35) is from CD4+ T cells, the authors only show data from B cells in Fig 7a. I therefore think it would be informative to show T cell data here as well as show that IL-12rb2KO controls the upregulation of mRNA for IL-10, p35 and Ebi3 by p35 (as shown in Fig 5b).

Reviewer #5 (Remarks to the Author):

This review only pertains to the use of the baculovirus-insect cell system and the expression of recombinant interleukin-12 (II-12p35) and the beta-chain subunit of Epstein-Barr virus induced host gene 3 (Ebi3) in insect (TnHigh-Five) cells. The transfer vector used, pMIB/V5-His (V8030-01), is a single entry vector, where the II-12p35 gene is expressed downstream immediate-early *Orgyia pseudotsugata* (OpMNPV) promoter ie-2. The *Autographa californica* nucleopolyhedrovirus (AcMNPV immediate early promoter ie-1 is driving the blasticin gene to select transformed cells. In my opinion both proteins express well and are appropriately secreted.

According to the submission the proteins secreted are partially dimers and partly monomers (Figure 1a), which is confirmed in Figure 1 b by Western analysis. I assume in both cases non-denaturing SDS-PAGE was used, but this is not clearly stated. If the authors used denaturing gels in case of Figure 1a and 1 b, there is a serious problem as the results are then inconsistent with Figure 1c and their preceding paper (Wang et al., 2014, Interleukin-35 induces regulatory B cells that suppress autoimmune disease, *Nature Medicine* 20: 646-641, Online Methods). The comments of authors to the reviewer comments seem to suggest denaturing gels were used.

In Figure 1b (Fractions 29-30) the position of the potential dimer position is not indicated, to show whether a fraction of the expressed II-12p35 is intrinsically dimer or monomer.

It is also clear from Figure 1b that there is monomer in the dimer fraction of the column. This may suggest that the dimers are quasi-stable. The presence of monomers in the dimer fraction and (possibly dimers in the monomer fraction may confound experiment described in Figure 3-5.

I assume the fractions 20-21 of the gel filtration experiment (Figure 1a) were used for sedimentation equilibrium experiments, but this is not clearly stated. If this is the case, the response of the authors on the oxidative state may (but with no references) be not relevant, when only the dimer fraction was used.

In conclusion. I think the experiments related to the expression of II-12p35 with Ebi3 are carried out correctly, but the description (also Materials & Methods) should be more detailed and unambiguous. See also comment #6 of the reviewer.

What is not (yet) clear to me is the issue of heterodimerization (II-12p35 with Ebi3) as brought up by the reviewer, as there is no mention in the current submission on the use of co-expressed II-12p35 with Ebi3. On this issue I tend to follow the authors.

Reviewer #4 (Remarks to the Author):**Summary**

This paper addresses whether the IL-35 subunit IL-12p35 possesses a previously unappreciated immune-regulatory function separate to its role in the formation of the IL-35 heterodimer IL-12p35/Ebi3. Although this study is extremely novel as (to my knowledge) this is a previously unexplored area of research as well as well-written and clearly explained there are some inconsistencies in the paper that need to be addressed.

Response:

We thank Reviewer for noting that our study is extremely novel and that the manuscript is well written and clearly explained.

Comment #1:

One inconsistency in the data is the interchangeability of the use of the p35 monomer and p35-p35 homodimer. Although the authors show that proliferation of CD4⁺ T cells anti-CD3/CD28 stimulated CD4⁺ T cells is similar following exposure to p35 monomer and p35-p35 homodimer they do not discount that these two forms have different effects on B cells (Fig. 1g only shows the effect of the monomer on B cells) or other effects in vivo. To address this the authors must show the same data in Fig. 3 for the p35 monomer and p35-p35 homodimer e.g. Day 21 is only shown for the fundus images for the p35-p35 treated mice and no ERG analysis is shown for p35-treated mice.

Response:

We felt it important to characterize and discuss the p35 monomer and p35-p35 homodimer because the insect cells expressing the IL-12p35 cDNA construct secreted both. After extensive characterization, we found no discernible difference between the p35 monomer and p35-p35 homodimer in vitro as well as in vivo. Data presented in Fig. 1 and Fig. 3 show that the p35 monomer and p35-p35 homodimer have similar effects on lymphocytes or during EAU. In response to the Reviewer's suggestion we have provided additional data (Fig. 1g; Fig. 3b; Fig. 3c; Fig.3d) that underscore the fact that both the monomer and homodimer have similar effects. In Fig.1g, the new data show that, similarly to CD4⁺ T cells, the p35 and p35-p35 have similar effects on B cells. In the EAU model, the ocular inflammation and pathological changes are traditionally characterized and diagnosed by funduscopy, histology or OCT. The new fundus (Fig. 3b), histology (Fig. 3c) and OCT data (Fig.3d) further confirm that the p35 and p35-p35 are both effective in mitigating EAU.

Comment #2

In fact, from the data that is shown it appears that p35 leads to a more significant

suppression of retinal inflammation than p35-p35 in treated mice. Indeed, it seems likely that as a monomer that p35 may more readily couple with Ebi3 *in vivo* increasing the amount of IL-35. With this in mind, it has also been previously shown that changing one subunit (e.g. IL-12p35) of the IL-12 family can change the balance between other family members (Vasconcellos et al., JI, 2011) and the authors do not measure IL-12, IL-27 or IL-23 levels follow *in vivo* administration with IL-12p35.

Response:

We agree with Reviewer that at first glance the data might suggest that p35 induces a more significant suppression of retinal inflammation than p35-p35. Moreover, p35 level increases during EAU while p35-p35 homodimer was not detectable (Fig. 2b), suggesting that p35 might be relevant in suppressing inflammation *in vivo*. However, we are reluctant to assert that p35 is a more effective Biologic for treatment of EAU because (i) our goal was not to make a direct comparison between p35 and p35-p35 homodimer and (ii) upon further evaluation we did not observe significant differences in their abilities to suppress EAU (Fig.3). In fact, both p35 and p35-p35 appear to be effective Biologics for treating Uveitis. We also agree with the Reviewer that changing one subunit (e.g. IL-12p35) of the IL-12 family can change the balance between other IL-12 family members and this has been a major challenge in understanding the physiological roles of the different IL-12 subunit proteins. As suggested by the Reviewer, one mechanism by which p35 exerts its immune suppressive effects might be to couple with Ebi3 *in vivo*, thereby increasing the amount of IL-35. In the future, we intend to directly determine the effects of IL-12p35 on IL-12, IL-27, IL-23 and IL-35 levels. In the revised manuscript, we have further discussed these aspects of the immunobiology of the IL-12 family and their subunits and have also added the citation to the 2011 JI manuscript by Vasconcellos et al.

Minor comment #1:

In the first section of the paper the authors describe the production of recombinant mouse p35 and Ebi3. However, in Fig. 1 there is only data that describes the size and conformation of p35 (monomer/homodimer) following analysis of the secreted peptides by SDS-page. Could the authors please explain why they have excluded the data concerning the size and conformation of recombinant Ebi3?

Response:

We did not present the data concerning the size and conformation of recombinant Ebi3 because the focus of the manuscript is to determine whether the IL-12p35 subunit can be developed as a Biologic to treat uveitis. In response to Reviewer's comment we have now provided the Ebi3 purification scheme, which provides this information (see Supplementary Fig.1).

Minor comment #2:

2. In reference to Fig. 1h the authors comment that ‘p35 treated cells displayed higher level of p27kip1....’ but in the figure, they only show the data from CD4+ T cells. Please change the text or add the B cell data to the figure.

Response:

In fact, the data (Fig.1h) is from CD19⁺B cells. Please, see Page 6, Lines 15-18.

Minor comment #3:

In Fig. 2 the authors show that splenocytes from LPS-injected mice express p35-p35 and Ebi3-Ebi3 homodimers but splenocytes from IRBP-immunized mice do not. The authors suggest that is due to the intense inflammatory conditions induced by LPS. Although I agree with the authors this is the most likely explanation, it could be due to the location of the cells draining the site of inflammation. With this in mind, it would be informative to also show data from cell lysates obtained from the LN draining the inflamed eye in Fig. 2b.

Response:

We agree with the Reviewer that the location of the cells draining the site of inflammation may in part account for the results. It is however of note that the data described in Fig. 2 was in response to a comment made by the previous Reviewer (Reviewer #1) who requested that we provide Western blot data showing that p35 and/or p35-p35 exist in vivo during an inflammatory response. As significant numbers of IL-35-producing regulatory B cells reside and traffic to the spleen during inflammation, it was logistically possible to perform the analysis using cell lysates obtained from the spleen. In contrast, the level of inflammatory cells or regulatory cells that traffic to the retina is relatively low compared to their levels in the periphery, making it technically challenging to detect p35 expression in the retina by Western blot analysis. Nonetheless, we believe that results of the Western blot analysis shown in Fig.2b reveal significant expression of p35 monomer during EAU, while the p35-p35 homodimer is detectable mainly during intense inflammatory immune responses, as may occur during sepsis. (Fig.2a).

Minor comment #4:

In Fig. 4a (Th1/Th17) and Fig. 4c (proliferation) the authors show T cell phenotype in the draining LN but in Fig. 4b (Foxp3) the authors showing T cell phenotype in the spleen. Please either show the data from the same organs, all the organs, or provide an explanation as to why different organs have been chosen.

Response:

We thank Reviewer for pointing out our mistake during manuscript preparation. We routinely isolate T cells from the LN and B cells from the spleen. The CD4 T

cells used for detecting regulatory T cells (Fig. 4b) were indeed from the LN and not the spleen. This error is now corrected in the revised manuscript. Please see page 9, lines 14-16; page 32, lines 13-16.

Minor comment #5:

Please show representative plots for Fig. 4d-f.

Response:

The representative plots for Fig. 4d-f are now provided in the revised manuscript.

Minor comment #6:

In Fig 1f the authors show that the stimulation of CD4+ T cells with CD3/CD28 in the presence of the 100ng/ml of p35 results in a four-fold decrease in proliferation. However, in Fig 5a this affect is dramatically less indeed the authors go on to comment in the text that Ebi3 is much more efficient at suppressing proliferation than p35. Although thymidine incorporation assays can be extremely variable this effect seems drastically altered between these two figures. As the authors use this data to conclude that Ebi3 has a different effect to p35 it is important for the authors to comment on this.

Response:

(i) We agree with the Reviewer that thymidine incorporation assays are extremely variable, particularly when the experiments are performed at different times and with different batches of Thymidine. This inherent variability of the thymidine incorporation assay is acknowledged in the revised manuscript (see page 10, lines 10-13).

(ii) We also agree with Reviewer that it would be interesting to look at the in vivo effects of Ebi3 on the development of autoimmune disease. In fact, we are currently investigating the in vivo effects of Ebi3 in autoimmune diseases, including EAU and EAE.

Minor comment #7:

Please show an isotype control for the gating of Ebi3/p35 double positive cells and IL-10 for the different B cell subsets in Fig. 5e.

Response:

Isotype control for gating of Ebi3/p35 and IL-10 are provided in the revised manuscript.

Minor comment #8:

In Fig. 6a a mixture of lymphocytes from the draining LN and spleen are restimulated to look at the production of P35+EBI3+ cells, could the authors please provide an explanation or show these data from these organs separately.

Response:

Local inflammatory diseases of the CNS, such as uveitis, are generally not as intense as systemic inflammation. Thus, in non-infectious uveitis it is generally acceptable to pool cells from the spleen and LN to obtain enough cells for analysis. This is particularly so for the analysis of regulatory cells that might comprise 1-4% of total B or T cells. In previous studies where we have used substantial numbers of mice to analyze regulatory B cells in these organs separately, we did not find significant differences between the two organs.

Minor comment #9:

Please show the summary data for Fig. 6c.

Response:

The summary data for Fig. 6c is provided in the revised manuscript.

Minor comment #10:

Although most of the proliferative data (Fig 1.f comparing monomer/homodimer, Fig. 5a comparing Ebi3, p35) is from CD4⁺ T cells, the authors only show data from B cells in Fig 7a. I therefore think it would be informative to show T cell data here as well as show that IL-12rb2KO controls the upregulation of mRNA for IL-10, p35 and Ebi3 by p35 (as shown in Fig 5b).

Response:

The data described in Fig. 7 was in response to the previous Reviewer (Reviewer #1) who requested that we provide evidence that; (i) p35-mediate biological effect of Breg cells is via IL-35 signaling pathway; (ii) that the loss of IL-35 signaling would inhibit p35-induced expansion of regulatory B cells. We believe that it is more appropriate to perform these studies using B cells because the IL-35-producing cells described in the manuscript are Breg cells. We therefore performed these experiments using B cells that lack IL-12R β 2, the quintessential receptor for IL-35 signaling. In line with data presented in Fig.1g, p35 suppressed the proliferation of the WT B cells while the loss of IL-12R β 2 signaling abrogated the inhibitory effect of p35 (Fig. 7a). Consistent with data shown in Fig.5e, stimulation of WT B cells with p35 induced expansion of IL-35-expressing B cells (Fig. 7b, 7c). In contrast, stimulation of IL-12R β 2 deficient B cells with p35 did not induce increase in the percentage of the IL-35-producing Breg cells (Fig. 7b, 7c). Taken together, these observations suggest that p35 requires IL-12R β 2 to mediate its anti-inflammatory activities.

Reviewer #5 (Remarks to the Author):

Summary

This review only pertains to the use of the baculovirus-insect cell system and the expression of recombinant interleukin-12 (Il-12p35) and the beta-chain subunit of

Epstein-Barr virus induced host gene 3 (Ebi3) in insect (TnHigh-Five) cells. The transfer vector used, pMIB/V5-His (V8030-01), is a single entry vector, where the Il-12p35 gene is expressed downstream immediate-early *Orgyia pseudotsugata* (OpMNPV) promoter ie-2. The *Autographa californica* nucleopolyhedrovirus (AcMNPV immediate early promoter ie-1 is driving the blasticin gene to select transformed cells. In my opinion both proteins express well and are appropriately secreted.

Response:

We thank Reviewer for noting that “both proteins express well and are appropriately secreted”.

Comment #1

According to the submission the proteins secreted are partially dimers and partly monomers (Figure 1a), which is confirmed in Figure 1 b by Western analysis. I assume in both cases non-denaturing SDS-PAGE was used, but this is not clearly stated. If the authors used denaturing gels in case of Figure 1a and 1 b, there is a serious problem as the results are then inconsistent with Figure 1c and their preceding paper (Wang et al., 2014, Interleukin-35 induces regulatory B cells that suppress autoimmune disease, *Nature Medicine* 20: 646-641, Online Methods). The comments of authors to the reviewer comments seem to suggest denaturing gels were used.

Response:

Non-denaturing (non-reducing) SDS-PAGE was used in the results described in Fig.1a and Fig.1b. We regret if our comments to the previous Reviewer #1 suggested that denaturing gels were used. We have clarified this point in the revised manuscript. Please see page 5, lines 16-23; page 6, lines 1-5).

Comment #2:

In Figure 1b (Fractions 29-30) the position of the potential dimer position is not indicated, to show whether a fraction of the expressed Il-12p35 is intrinsically dimer or monomer.

Response:

We have revised Fig. 1b and have indicated the position of the p35 monomer and p35-p35 homodimer in the Figures.

Comment #3:

It is also clear from Figure 1b that there is monomer in the dimer fraction of the column. This may suggest that the dimers are quasi-stable. The presence of monomers in the dimer fraction and (possibly dimers in the monomer fraction may confound experiment described in Figure 3-5.

Response:

We agree with reviewer that because the p35 subunits that make-up the p35-p35 homodimer are not covalently linked, presence of monomers in the dimer fraction may occur. This consideration was the reason for using Amicon Ultra centrifugal filter units to separate out the monomers from the dimers and further subjecting the preparations to several cycles of FPLC purification to obtain the highly purified p35 monomer independent of the p35-p35 homodimer. Nevertheless, in both in-vitro and in-vivo studies, we did not find discernible difference between the effects of p35 and the p35-p35 homodimer; In our opinion, this obviated any concern that trace amounts of either protein in the p35 or p35-p35 preparation might confound interpretation of our results. Moreover, our goal was to establish that IL-12p35 (p35 or p35-p35) could serve as effective Biologic to treat Uveitis.

Comment #3:

I assume the fractions 20-21 of the gel filtration experiment (Figure 1a) were used for sedimentation equilibrium experiments, but this is not clearly stated. If this is the case, the response of the authors on the oxidative state may (but with no references) be not relevant, when only the dimer fraction was used.

Response:

Fraction 20-21 was subjected to two cycles of FPLC purification and enrichment with Amicon Ultra centrifugal size exclusion filters before use for sedimentation equilibrium. This point is now stated in the revised manuscript. Please see page 6, lines 1-3.

Comment #4:

In conclusion. I think the experiments related to the expression of IL-12p35 with Ebi3 are carried out correctly, but the description (also Materials & Methods) should be more detailed and unambiguous. See also comment #6 of the reviewer.

Response:

We thank the Reviewer for noting that the experiments related to the expression of IL-12p35 or Ebi3 are carried out correctly. We have followed Reviewer's advice and paid particular attention to the Materials & Methods section. Indeed, the description of the methodology in the revised manuscript is more detailed and unambiguous. Please see page 5, lines 16-23; page 6, lines 1-5; page 18, lines 7-24; page 30, lines 6-19.

Comment #5:

What is not (yet) clear to me is the issue of heterodimerization (IL-12p35 with Ebi3) as brought up by the reviewer, as there is no mention in the current submission on the use of co-expressed IL-12p35 with Ebi3. On this issue I tend to follow the authors.

Response:

We fully agree with Reviewer as the current study under review relates to the investigation of whether the IL-12p35 subunit possesses immune-suppressive actions that can be exploited for the treatment of uveitis.

Reviewers' comments:

Reviewer #4 (Remarks to the Author):

The authors have addressed all my questions.

However, the IL-10 staining is not convincing so it is important if they could confirm the increase in IL-10 by ELISA/PCR following treatment in vivo (like they did in vitro).

The authors should also show what happens to the frequency of T2-MZPs and B10 following treatment as they look at B cell subset based on CD38/CD138/IgG2a/IgD.

Minor comments: there is a typo in figure 4e – the authors show a CD11b/CD45 on the facs plot but then show the bar charts for CD45+F4/80 cells.

Reviewer #4 (Remarks to the Author):

Summary

The authors have addressed all my questions.

Response:

We thank Reviewer for noting that we have addressed all the questions raised.

Comment #1:

The IL-10 staining is not convincing so it is important if they could confirm the increase in IL-10 by ELISA/PCR following treatment in vivo (like they did in vitro).

Response:

In response to Reviewer's comment we have provided new data that further validate our FACS data showing that p35 can induce the expansion of IL-10 producing regulatory T cells. The new RT-PCR data shows increase in IL-10 mRNA transcripts (Fig. 4c) and the ELISA data (Fig. 4d) shows increase of IL-10 secretion in LN of mice treated with p35 compared to untreated mice.

Comment #2:

The authors should also show what happens to the frequency of T2-MZPs and B10 following treatment as they look at B cell subset based on CD38/CD138/ IgG2a/IgD.

Response:

This study demonstrates that p35 can induce the expansion of IL-10- and IL-35-expressing regulatory B cells (Bregs) and can ameliorate autoimmune uveitis. However, examination of the various Breg types or sites or that harbor the regulatory cells is beyond the scope of the current study. Although we analyzed the frequency of T2-MZPs and B10 following p35 treatment, we do not believe that this observation is relevant to our conclusions and may give the impression that all the Breg cells that mitigate uveitis are T2-MZPs and B10. We are therefore reluctant to include this incomplete data in the manuscript. Nonetheless, we have provided this preliminary data for review purpose only. Please see data designated as "For review purpose only."

Minor comment:

Minor comments: there is a typo in figure 4e – the authors show a CD11b/CD45 on the FACS plot but then show the bar charts for CD45+F4/80 cells.

Response:

We thank the Reviewer for pointing out this error. The error has been corrected in the revised manuscript. Please see revised Fig. 4g).

Note: All changes made to the text are highlighted.